# Visualizing ultrafast photothermal dynamics with decoupled optical force nanoscopy

Hanwei Wang [1,2], Sean M. Meyer[3], Catherine J. Murphy [3], Yun-Sheng Chen [1,2,4,5,6] & Yang Zhao [1,2,4,5,7] ✉

The photothermal effect in nanomaterials, resulting from resonant optical absorption, finds wide applications in biomedicine, cancer therapy, and microscopy. Despite its prevalence, the photothermal effect in light-absorbing nanoparticles has typically been assessed using bulk measurements, neglecting near-field effects. Beyond standard imaging and therapeutic uses, nanosecond-transient photothermal effects have been harnessed for bacterial inactivation, neural stimulation, drug delivery, and chemical synthesis. While scanning probe microscopy and electron microscopy offer single-particle imaging of photothermal fields, their slow speed limits observations to milliseconds or seconds, preventing nanoscale dynamic investigations. Here, we introduce decoupled optical force nanoscopy (Dofn), enabling nanometer-scale mapping of photothermal forces by exploiting unique phase responses to temporal modulation. We employ the photothermal effect's back-action to distinguish various time frames within a modulation period. This allows us to capture the dynamic photothermal process of a single gold nanorod in the nanosecond range, providing insights into non-stationary thermal diffusion at the nanoscale.

Optical forces, also known as light-induced forces, refer to the mechanical effects generated by the interactions between light and matter. When light is absorbed or scattered by a material, it can create non-uniform distributions of the electromagnetic or thermal fields, resulting in mechanical forces due to the transfer of momentum. These optical forces have been widely used in a variety of applications, including trapping[1,2], sensing[3], micromanipulation[4,5], and surface characterization[6].

The measurements of microscopic optical forces have been accomplished using techniques such as atomic force microscopy (AFM)[7,8], photo-induced force microscopy (PiFM)[6,9,10], and far-field scattering methods[11,12]. However, the interpretation of these measurements has been complicated by the fact that optical force is an

umbrella term encompassing a wide range of forces generated by light-matter interactions. Decoupling these forces is crucial for understanding the optical, thermal, and mechanical properties of materials and for developing efficient sensing, imaging, trapping, and actuating schemes. Despite its importance, decoupling optical forces remains a significant challenge due to the complex nature of light-matter interactions. New techniques and theoretical models that accurately measure and decouple different types of optical forces could lead to advancements in our understanding of light-matter interactions and the development of applications in a wide range of fields as diverse as nanophotonics, biophysics, and materials science.

While scanning probe-based force measurement generally provides a high spatial resolution of optical forces; three forces—optical

[1]Department of Electrical and Computer Engineering, University of Illinois Urbana-Champaign, Urbana, IL, USA. [2]Nick Holonyak Micro and Nanotechnology Laboratory, University of Illinois Urbana-Champaign, Urbana, IL, USA. [3]Department of Chemistry, University of Illinois Urbana-Champaign, Urbana, IL, USA. [4]Beckman Institute for Advanced Science and Technology, University of Illinois Urbana-Champaign, Urbana, IL, USA. [5]Department of Bioengineering, University of Illinois Urbana-Champaign, Urbana, IL, USA. [6]Department of Biomedical and Translational Sciences, Carle Illinois College of Medicine, University of Illinois Urbana-Champaign, Urbana, IL, USA. [7]Carl R. Woese Institute of Genomic Biology, University of Illinois Urbana-Champaign, Urbana, IL, USA. ✉e-mail: yzhaoui@illinois.edu

gradient force, photothermal force, and photoacoustic force—are tangled and measured by the probe simultaneously. An approach using PiFM to decouple the optical gradient force from the photothermal expansion using various thicknesses of polymer films is documented[13]; however, it relies on sample thickness and is not applicable to nanomaterials or nanostructures. Additionally, in some cases, optical forces can be transient but have not been measured directly at the nanoscale. The dynamic information could be lost due to the relatively slow scanning speed of the probe. Although improved mechanical designs of the scanner have enabled a high-speed atomic force microscope (Hs-AFM) that can reach up to 1300 frames per second, a single image-frame still takes hundreds of microseconds, significantly longer than the thermal relaxation time of nanoparticles, which is in the nanosecond regime.

Here, we develop a decoupled optical force nanoscopy (Dofn) that can map the optical forces, capitalizing on the unique phase responses of the different optical force components under a specific temporal modulation profile of light. We measured the spatial distribution of these piconewton-level optical force components generated from a single gold nanoparticle with 10 nm resolution. We further demonstrate an ultrafast visualization of dynamic heat transfer in the nanosecond temporal regime using the back-action effect[14]. We show the heating and cooling stages of the gold nanoparticle using Dofn. Our method provides a promising solution to the long-standing challenge of measuring the fast dynamics of force evolution at the nanoscale.

## Results

### The decoupled optical force nanoscopy (Dofn) system setup

Dofn is built upon an inverted optical microscope that allows modulated and polarization-controlled illumination of the sample from the bottom and is integrated with an atomic force microscope (AFM) that probes the sample atop. While conventional near-field scanning optical microscopy (NSOM)[15] can also probe the near-field electromagnetic distributions around a nanoscale sample but not specifically the photothermal forces because NSOM relies on the scattered light that is primarily related to the radiation pressure. Our technique, on the other hand, probes optical forces originating from the interactions between the illuminated nanoparticle and the probe. Such optical forces carry rich information about the nanoparticle including its thermal properties but also involve complex interplay among the different types of forces generated simultaneously by light. Depending on the material properties of the nanoparticle, there are primarily three types of optical forces: the photothermal force, photoacoustic force, and radiation pressure (positive due to forward scattering and negative due to optical gradient[13], with the first and second forces related to the thermal properties of the nanoparticle. The photothermal force[10,16] is on a similar magnitude scale as the radiation pressure and, therefore, cannot be easily delineated from the rest of the optical forces[7,9,17]. In the following, we show that Dofn decouples the photothermal force from the optical gradient force and the non-localized forces (the photoacoustic force as well as the scattering force), capitalizing on their unique phase responses under a specific waveform of the optical excitation.

Figure 1a (and Supplementary Fig. 1) shows the setup of the Dofn system. A supercontinuum laser acts as a continuous wave (CW) source that is temporally modulated by a square wave with a modulation frequency near the fundamental mechanical resonance of the AFM probe in the hundred-kilohertz regime, $f_{opt}$. At the same time, the piezo on the AFM probe mechanically dithers the cantilever at a slightly different frequency, $f_d$. As the optical forces and the mechanical dithering force from the piezo shake the probe at different frequencies, the deflection signal from the cantilever recorded by the built-in photodetector contains both frequency components. The vibration at $f_d$ indicates the topography of the nanoparticle, and that

at $f_{opt}$ indicates the optical forces from the nanoparticle upon illumination (Fig. 1b). The optical forces are extracted with a lock-in amplifier with a corresponding reference frequency of $f_{opt}$.

### Physical origins of the optical forces

We choose gold nanorods[18] as our nanoscale sample because of their promising applications in photothermal therapy of cancers[19,20] as well as the complex optical forces involved. The nanorods are chemically synthesized and capped with polyethylene glycol (PEG). The nanorods have a dimension of around 90 nm in the longitudinal axis and around 30 nm in the transverse axis that resonate at around 700 nm (in the air on a dielectric substrate). We drop cast the gold nanorods on a polymethyl methacrylate (PMMA) substrate which has similar thermal properties to adipose tissue[21] (Supplementary Fig. 2). The PEG coating on the nanorod mimics the real-case applications for their use as biocompatible photoacoustic imaging contrast agents[19,22] or photothermal therapeutic agents[19,20].

Dofn measures optical forces that are collectively attributed to the photothermal effect, the photoacoustic effect, and the optical gradient effect (negative radiation pressure). The positive radiation pressure generated from the scattered light from the gold nanorod is relatively small compared to the other optical force components as the absorption dominates the total extinction in small particles[23]. The optical gradient force[24], $F_G$, is associated with the electric polarizabilities of the AFM probe, $\alpha_{tip}$, as well as the electric field intensity in the z-direction at the AFM tip location, $E_z$ (Fig. 1c, Supplementary Note 5).

$$F_G \propto \alpha_{tip} \nabla E_z{}^2 \tag{1}$$

Although the AFM probe is significantly larger than the wavelength, the interacting region is limited to the tip of the AFM probe, as shown in Supplementary Fig. 3 (Supplementary Note 2). Therefore, the relationship for small particles, such as described in Ref. 23, still holds true. The radiation pressure is independent of the engagement condition. It is removed at the beginning as a constant background force.

The photothermal force, $F_{PT}$, is proportional to the overall thermal expansion of the nanorod and the AFM probe around the nanoparticle-tip interface (Fig. 1c and Supplementary Fig. 3).

$$F_{PT} \propto \int_{probe} dz \beta \Delta T \tag{2}$$

where $\beta$ is the thermal expansion coefficient, and $\Delta T$ denotes the elevated temperature of each domain, including the nanorod and its surrounding media with an assumption that only the nanorod absorbs optically (Supplementary Note 2). When the nanorod is illuminated at its plasmonic resonance, it also generates a photoacoustic signal with the modulation frequency of the laser, $f_{opt}$. The acoustic pressure is proportional to the second time derivative of the photothermal expansion[25].

$$F_{PA} \propto \int_{probe} ds \beta \frac{d^2 T}{dt^2} \tag{3}$$

As the photoacoustic pressure is generated by the second time derivative of the temperature, the photoacoustic pulse width, as shown in Fig. 1d, is around 471 ns. The pulse length in air is, therefore, around 160 μm. It is on the same length scale as the cantilever's size of around 175 μm but much larger than the tip diameter of around 20 nm. As a result, the photoacoustic pressure will exert on the entire probe (i.e., cantilever and tip) instead of a localized spot near the tip, resulting in a non-localized photoacoustic force (Supplementary Note 4), which can be treated as a uniform background.

## Differential phase responses of optical forces

It is important to note that the excitation laser should be modulated with a square waveform because the three optical forces will show distinguishable temporal responses and lead to different phases in the frequency domain. The optical gradient force responds to the changes of the laser intensity in a time scale of femtoseconds, which is considered to be instantaneous[26]. As shown in Fig. 1d, the optical gradient force is an attractive force proportional to the modulation waveform because it is generated by the electric field gradient near the gold nanorod. The photothermal force has a 282.9 ns time delay in response to the laser's intensity change due to the heat capacitance of the nanorod and its surrounding media based on our numerical calculation using a finite element solver (COMSOL Multiphysics 5.5). The photoacoustic force follows the second time derivative of temperature changes, one should expect two opposite peaks at the rising and falling edges during each modulation cycle of the excitation laser. The photoacoustic signal from a nanorod is thermally non-confined, which means that the absorbed heat of the nanorod diffuses into the media

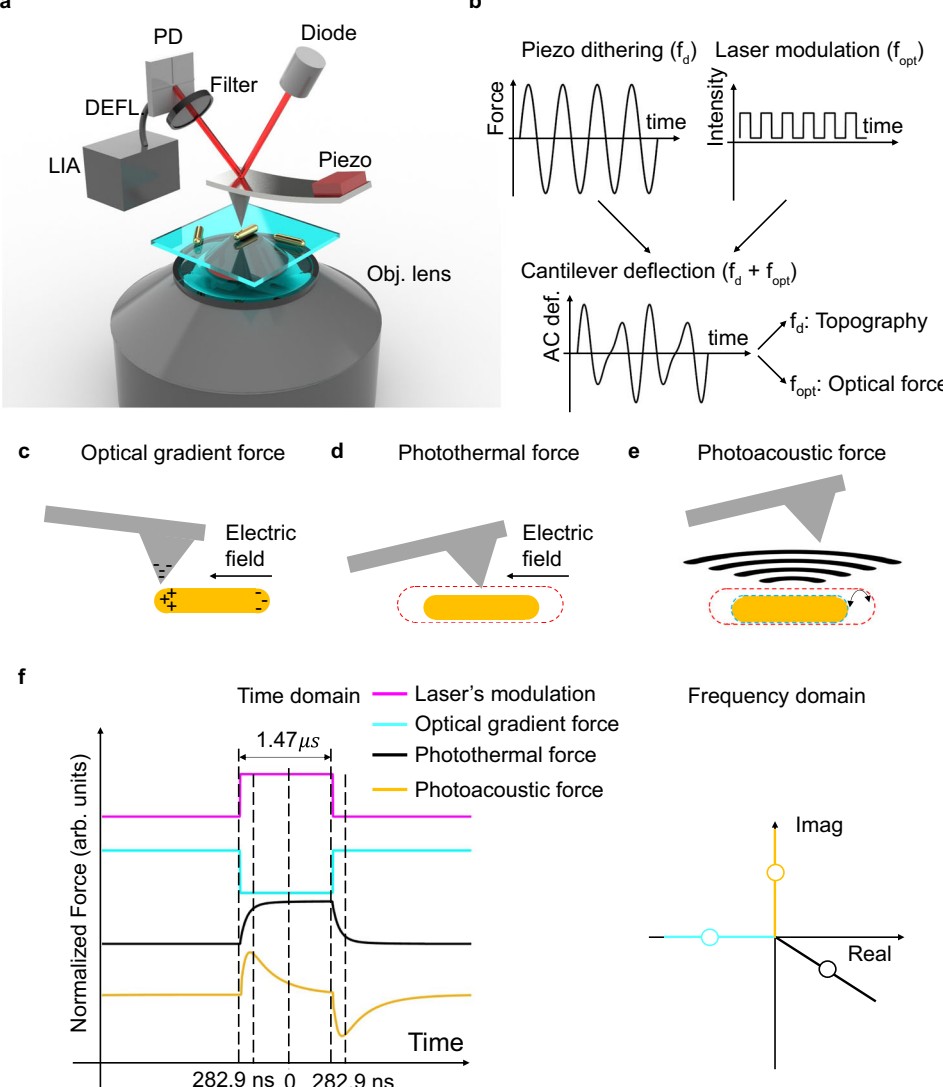

**Fig. 1 | Operational schematics of decoupled optical force nanoscopy (Dofn).**
**a** The schematic of the Dofn system. A modulated laser is focused on the sample through an objective lens. The AFM tip measures the optical forces on the transmission side. The deflection signal are demodulated with a lock-in amplifier with the reference frequency of $f_{opt}$. DEFL: deflection signal; LIA: lock-in amplifier; PD: photodetector. **b** Illustration of the piezo's modulation signal, the laser's modulation signal, and the resulting cantilever deflection in the time domain. The piezo is driven by a sinusoidal wave with the frequency of $f_d$ and the laser is modulated by a square wave with the frequency of $f_{opt}$. The deflection signal at $f_d$ contains topographic information and at $f_{opt}$ contains the optical force information. Schematic illustration of (**c**) optical gradient force, (**d**) photothermal force, and (**e**) photoacoustic force. **f** Simulated optical forces from a nanorod on PMMA substrate measured by an AFM probe in the time domain and illustrated corresponding optical forces in the frequency domain, showing distinct phase distributions of the gradient force ($F_G$, cyan), photothermal force ($F_{PT}$, black), and photoacoustic force

($F_{PA}$, orange). In the time domain, the optical gradient force simultaneously responds to the laser excitation and follows the temporal modulation (magenta). The photothermal force has around 283 ns delay to reach 90% of its stationary intensity. The photoacoustic force only happens at the rising and falling edges and has a full width at half maximum of around 471 ns. In the frequency domain, the optical gradient force is negative and pure real; the photothermal force is complex; and the photoacoustic force is positive and pure imaginary. The insets illustrate the physical mechanisms of each force. When the gold nanorod is on plasmonic resonance, the electric field gradient at the two ends of the nanorod will induce a polarization of the AFM tip, which generates the optical gradient force. The thermal expansion from the generated heat due to the optical absorption of the nanorod causes the photothermal force. The oscillating thermal expansion generates the acoustic wave and the acoustic pressure acting on the AFM probe, resulting in the photoacoustic force.

and the photoacoustic signal is largely contributed by the immediate media surrounding the nanorod[27]. These distinguishable temporal responses lead to different phases of the optical forces. The optical gradient force is an even symmetric function with respect to the center of the modulation period, and therefore, falls on the real axis in the frequency domain. The photothermal force has both even and odd symmetric components, and therefore, is a complex value in the frequency domain. The photoacoustic force is an odd symmetric function in the time domain, and thus, falls on the imaginary axis in the frequency domain. Based on these known distributions in the frequency domain, we can therefore distinguish the different origins of optical forces using their unique phases.

Figure 2 shows the measured topography (Fig. 2a) and the complex optical forces (both amplitude and phase in Fig. 2b, c, respectively) of a gold nanorod through a single scan. The three types of optical forces have not only distinct phases but also different spatial distributions. Figure 2c shows the highly structured phase distributions of the optical forces. In particular, the phase is close to −180 degrees around the two ends of the nanorod, which suggests that the optical gradient effect dominates at the ends of the nanorods. The phase is between 0 and 90 degrees over the body of the nanorod, suggesting that the photothermal effect dominates in that region. In contrast, the phase is close to 90 degrees in most of the background region, which corroborates that the non-localized photoacoustic effect dominates the background signal but contributes minimally to the structured distribution around the gold nanorod.

We define the amplitude distribution of the optical forces as $F_{opt}(\vec{r})$ and phase as $\varphi_{opt}(\vec{r})$. To delineate these optical forces, we first subtract the non-localized photoacoustic force $F_{PA}$ and its phase $\varphi_{PA}$ in the field of view based on the background signal outside the nanorod. We assume the background is mainly from the photoacoustic force, and thus it has a phase of 90 degrees. We find the phase of the photothermal force $\varphi$ from the region with a low contribution of the optical gradient force (i.e., the center of the nanorod). When the photoacoustic force is removed from the overall optical forces, the photothermal force is the only one that contributes to the imaginary component of the remaining optical forces$(F_{opt}(\vec{r}) - F_{PA}) \sin(\varphi_{opt}(\vec{r}) - \varphi_{PA} + \pi/2)$. Thus, the photothermal force can be calculated as $F_{PT}(\vec{r}) = \frac{(F_{opt}(\vec{r}) - F_{PA}) \sin(\varphi_{opt}(\vec{r}) - \varphi_{PA} + \pi/2)}{\sin(\varphi_{PT})}$. The optical gradient force only has a real component in our hypothesis and is decoupled by subtracting the real component of the photothermal force, $F_G(\vec{r}) = (F_{opt}(\vec{r}) - F_{PA}) \cos(\varphi_{opt}(\vec{r}) - \varphi_{PA} + \pi) - F_{PT}(\vec{r}) \cos(\varphi_{PT})$. The maps of the photothermal force and the optical gradient force are shown in Fig. 2d–f and confirmed by the simulated force maps in Fig. 2g–i. Using the same procedure, we decoupled the optical gradient forces with different circular polarizations at various wavelengths, showing the distinct force distributions around the nanorod that corroborates with electromagnetic simulations (Supplementary Fig. 5). The spectrum of the optical force at different probing locations is shown in Fig. 2j and can fit well with the simulation (Fig. 2k). We observed the plasmonic resonance around the wavelength of 700 nm as expected, which matches the resonance of the photothermal force and the optical gradient force. The resonance peak measured at the end of the nanorod is red-shifted by around 10 nm from that of the optical gradient force measured at the center of the nanorod. This shift may attribute to the near-field coupling between the AFM probe and the nanorod[28].

## Photothermal back-action for time selectivity

A substantial challenge of scanning probe techniques is to capture the temporal dynamics of the photothermal process because, intuitively, the mechanical oscillation and the scanning speed of a probe are much slower than the photothermal dynamics, which is in the nanosecond to microsecond regime depending on the substrate. To freeze the time, we utilize the property that the photothermal

force is amplified when the AFM tip is in close contact with the sample.

The photothermal expansion, $B = \beta \Delta T$, drives a periodic deflection signal at $f_{opt}$. The oscillation of the photothermal expansion results in a time-varying spring constant $k_{PT}(t)$, which is stronger when the tip is in contact with the sample and vice versa. The probed photothermal force is proportional to this spring constant, $F_{PT} \propto \langle e^{i\omega_{opt}t} | k_{PT}(t) B(t) \rangle$. The oscillation of the spring constant results in a distinct response of different time frames within a period. The time frame when the tip is in close contact with the sample is selectively enhanced. This phenomenon is known as the back-action of the photothermal expansion[14]. As shown in Fig. 3a, the tuning curve of the cantilever shifts at different laser intensities (Supplementary Note 6). The tip-sample interaction is modulated by the laser's intensity and oscillates at $f_{opt}$ with a phase shift of $\phi$ to the photothermal expansion. The cantilever shows a higher response to the time frame in a modulation period when it has a higher tip-sample interaction. Therefore, the detected photothermal expansion is locked at a certain time frame within a period. The probed time frame can be tuned by the phase shifts between the photothermal expansion and the tip-sample interaction. We model the Dofn system with the following time-domain equation:

$$m\ddot{z} + m\gamma\dot{z} + kz = F_d + F_{opt} + F_{int} \qquad (4)$$

where $m$ is the effective mass of the probe, $z$ is the deflection of the cantilever, $\gamma$ is the damping coefficient, which comes from the viscosity of the sample and the air resistance to the probe and is assumed to be a constant. $k$ is the spring constant of the probe. $F_d$ is the dithering force of the piezo, which has a frequency dependence of $\omega_d$, $\omega_d = 2\pi f_d$. $F_{opt}$ denotes the overall optical forces. $F_{int}$ describes the tip-sample interaction force that occurs when the tip engages to the surface when the tip is oscillating. Here, we assume a linear tip-sample force, $F_{int} = -k_{sp}z$, where $k_{sp}$ describes an effective spring constant in addition to the free air spring constant of the probe, and it is determined by the engagement factor $\xi$. When solving Eq. (4) for the cantilever deflection, it has two solutions at the optical modulation frequency— one corresponds to the attractive mode and the other corresponds to the repulsive mode (Supplementary Note 7):

$$\langle e^{i\omega_{opt}t} | Z_{opt}^{\pm} \rangle = \frac{F_{opt}}{-m\omega_{opt}^2 + im\gamma\omega_{opt} + m\omega_d^2 \pm \frac{1}{\xi}\sqrt{(k - m\omega_d^2)^2 + (1 - \xi^2)(m\gamma\omega_d)^2}} \qquad (5)$$

where $|Z_{opt}^{+}\rangle$ denotes the deflection of the repulsive mode and $|Z_{opt}^{-}\rangle$ for the attractive mode. In the experiment, we use a 60% amplitude as the setpoint of the engagement, corresponding to an engagement factor $\xi$ of 0.6. Our probe has a mechanical resonance frequency of 174.1 kHz, a spring constant of 8.6 N/m, and a damping coefficient of $5.65 \times 10^3 \text{s}^{-1}$, which are measured quantities. In repulsive or attractive modes, we plot the theoretical frequency-dependent map of the amplitude and phase of the optical forces, as a function of the relative frequency with respect to the cantilever mechanical resonance (Fig. 3b–d). As each AFM probe has a slightly different fundamental mechanical resonance, $f_0$, we use relative frequencies with respect to the mechanical resonance to calibrate the system. The two modes are observed experimentally (Fig. 3e–g), which corroborate our theory. As the attractive mode is in a lower energy state, a stronger signal intensity can be observed; therefore, we choose to measure optical forces on this mode.

The probed time frame can be tuned by the phase of the cantilever deflection, $\phi$, as $t_{probe} = t_0 + \frac{\phi}{f_{opt}}$, where $t_0$ is a constant given by the initial condition. The phase $\phi$ is dependent on both the optical modulation frequency and the piezo's dithering frequency (Fig. 3c, f). In our measurement, we fix the optical modulation frequency close to

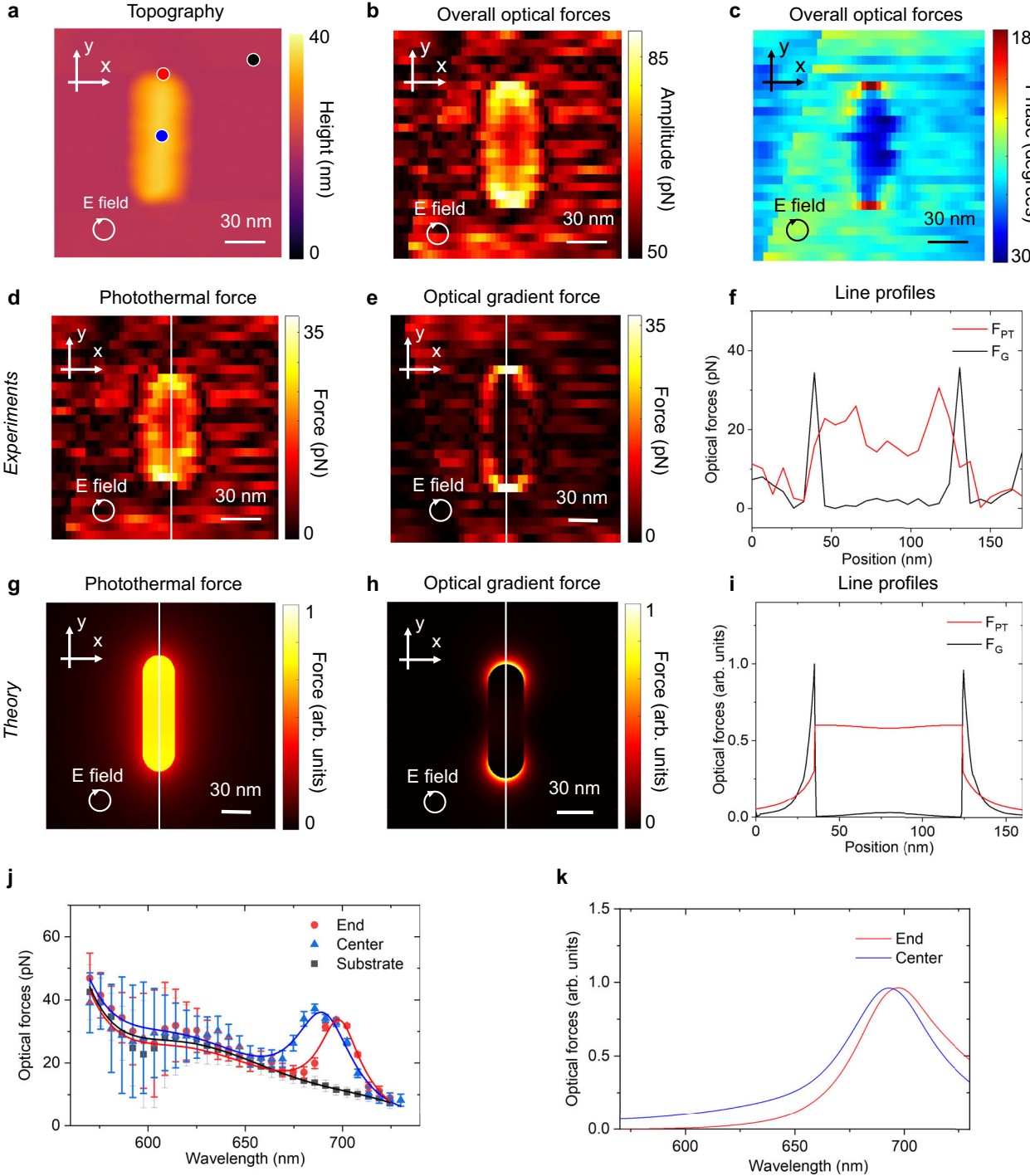

**Fig. 2 | Decoupled optical forces from different physical mechanisms. a** AFM topography of a nanorod. **b** Overall force map of the nanorod, $F_{opt}(\vec{r})$. The illumination is left-handed circularly polarized. The field of view is 150 nm by 150 nm. **c** The phase of the optical force with the background phase correction $\varphi_{opt}(\vec{r}) - \varphi_{PA} + 90°$. The optical gradient force, $F_G(\vec{r})$, is dominant at two ends of the nanorod, where the phase is close to 180 degrees. The photothermal force, $F_{PT}(\vec{r})$, is dominant at the body of the nanorod, where the phase is between 0 and 90 degrees. The photoacoustic force is dominant outside of the nanorod, where the phase is around 90 degrees. The decoupled amplitude of the measured (**d**) photothermal force and (**e**) the optical gradient force. **f** Measured decoupled optical gradient force and photothermal force along the dashed line as shown in (**d**)

and (**e**). Simulated (**g**) photothermal force map and (**h**) optical gradient force map at 100 ns after the raising edge. **i** The line profile of simulated optical gradient and photothermal forces along the dashed line as shown in (**g**) and (**h**). **j** Measured optical force spectra on the substrate (black), at the end (red), and the center (blue) of the nanorod, as indicated in (**a**). The laser modulation, $f_{opt}$, is at the same frequency as the mechanical resonance frequency of the cantilever, $f_0$. The frequency difference $f_d - f_0$ is 1 kHz. The error bars indicate the standard deviation, and the data points indicate the mean values of three measurements. **k** The simulated optical force spectra at the end and the center of the nanorod are given by the optical gradient force and the photothermal force, respectively.

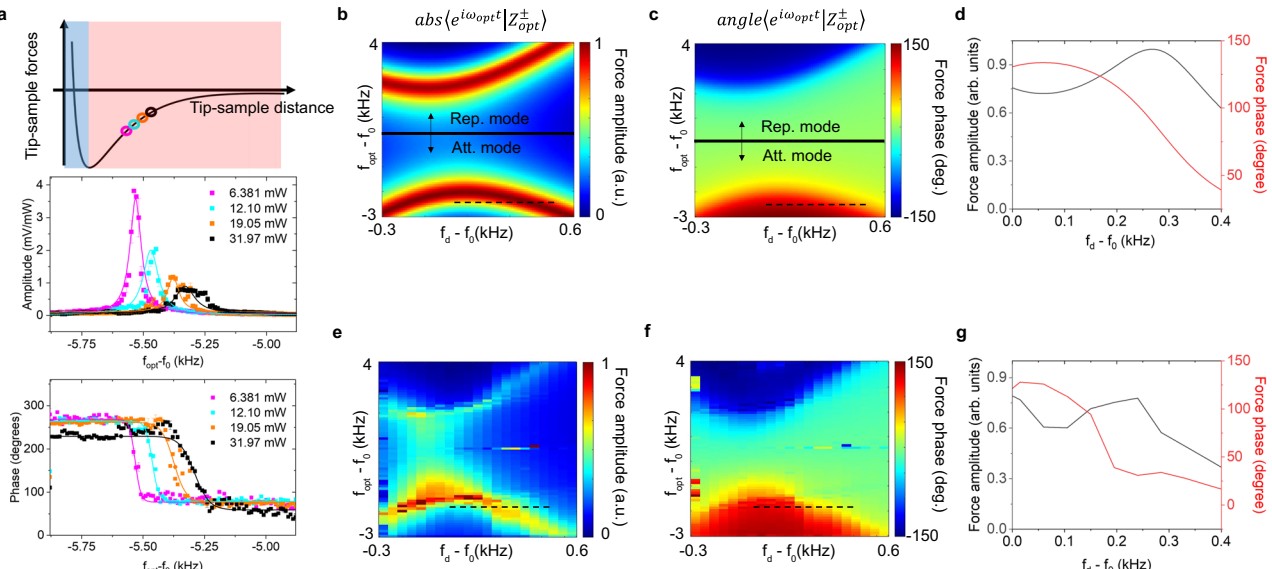

**Fig. 3 | Time-selective photothermal measurements. a** Top: schematics of the tip-sample interaction with the repulsive mode (blue) and the attractive mode (red) regimes. Amplitude (middle) and phase (bottom) of the measured tuning curve of the cantilever at different laser's intensity. The resonance is shifted due to the back action of the photothermal expansion. The error bars indicate the standard deviation, and the data points indicate the mean values of three measurements. **b** Theoretical magnitude of the repulsive and attractive $Z_{opt}^{\pm}$ modes. Norm. Amp.:

Normalized amplitude. $f_0$ is the mechanical resonance frequency of the cantilever, $f_d$ is the piezo dithering frequency, and $f_{opt}$ is the laser frequency. **c** Theoretical phase of the $Z_{opt}^{\pm}$ modes. **d** Amplitude and phase as shown in (**b**) and (**c**). **e** Measured magnitude of the cantilever's deflection at $f_{opt}$ versus piezo's frequency and optical frequency. The dashed line is at $f_{opt} - f_0$ of −1.5 kHz. **f** Measured phase of the deflection at $f_{opt}$. **g** Measured amplitude and phase at the line shown in (**e**) and (**f**).

the upper edge of the $|Z_{opt}^{-}\rangle$ mode to ensure a relatively large signal to noise ratio and a close-to-constant amplitude; by varying the piezo's dithering frequency, as shown in the dashed lines in Fig. 3d, e, the phase response of the probe at $f_{opt}$ can be tuned. We fix the relative optical modulation frequency at −1.38 kHz on the attractive mode branch and alter the relative piezo's dithering frequency from −0.75 kHz to 0.75 kHz, which corresponds to tuning the phase from around 0 to around 150 degrees as indicated by the dashed line in Fig. 3f.

By varying the piezo's dithering frequency, we tune the phase of the cantilever's deflection with respect to the laser's modulation, and therefore, observe the heating and cooling stages of a single nanorod within one modulation period. The temporal resolution of the measurement is associated with the phase tuning resolution. Based on the phase jittering limited by the fluctuation of the AFM system, the temporal resolution is estimated to be 32.7 ns (Supplementary Note 8).

We use a PMMA substrate, which has a higher heat capacitance and lower thermal conductivity than a glass substrate to better mimic the heat transfer of nanoparticles inside adipose tissue. The low thermal conductivity also prolongs the heat transfer process. As shown in Fig. 4a, the thermal relaxation time is 282.9 ns, thus, the heating and the cooling processes take a significant proportion (around 38%) of the modulation period. The process of heat conduction is non-stationary. As shown in the theoretical calculation (Fig. 4b–f), the nanorod, which resonantly absorbs photon energy and generates heat, is the first region to heat up. As the thermal gradient forms between the nanorod and the surrounding media, heat diffuses out and heats up the surrounding region of PMMA in 282.9 ns, which is the nanoscopic expression of the thermal non-confinement. As the gold nanorod has a better thermal conductance to the AFM tip compared to PMMA, the equilibrium thermal expansion of the surrounding PMMA is higher than the gold nanorod (Fig. 4c, d). Similarly, during the cooling phase, the nanorod will first cool down due to the high thermal conductance to the AFM tip (Fig. 4e) and followed by the surrounding media (Fig. 4f). The measured time dynamics of the photothermal force maps in Fig. 4g–k match well with the theoretical prediction (Fig. 4b–f).

## Discussion

We have demonstrated a technique for experimentally measuring optical forces with various physical origins in a time scale of nanoseconds and a spatial resolution of nanometers. Using the decoupled force nanoscopy (Dofn), we show the transient photothermal responses from a gold nanoparticle and its heating and cooling processes with distribution in two dimensions. Our experiments also demonstrate the spectroscopic dependence of the optical forces at various locations from the nanorods with different force dominance, confirming that the optical force arising from the photothermal expansion has a distinct phase profile compared to that from the radiation pressure.

While our measurements show a temporal resolution in hundreds of nanoseconds with a limitation of a few tens of nanoseconds because we choose to study biomedical relevant materials that require ambient conditions, a higher sensitivity of the measurement can be achieved by reducing the thermal noise and sideband leakage, for example, by proceeding with measurement in ultra-high vacuum and low temperature. This technology can be extended to any sample that follows three criteria: first, the sample is compatible with AFM; second, the sample has a strong interaction with light; and third, to measure the photothermal force temporal profile, the thermal relaxation is in a time scale comparable with the cantilever's mechanical resonance. Examples of such samples include polymers, 2D materials, nanoparticles, cells, and macromolecules. However, since the time-resolved capability of this technique requires repetitive modulation of the laser, it cannot observe single-shot or random photothermal processes. Additionally, our measured dynamic signals are related to photothermal expansion, and they do not intend to replace traditional pump-probe techniques that target ultrafast processes related to general light-matter interactions[29].

Our technique adeptly decouples various optical force components and captures the temporal dynamics of photothermal forces. Measuring nanoparticles paves the way for crafting effective photothermal nano-agents. For instance, by tweaking the nanoparticle's material composition, we can discern distinct photothermal

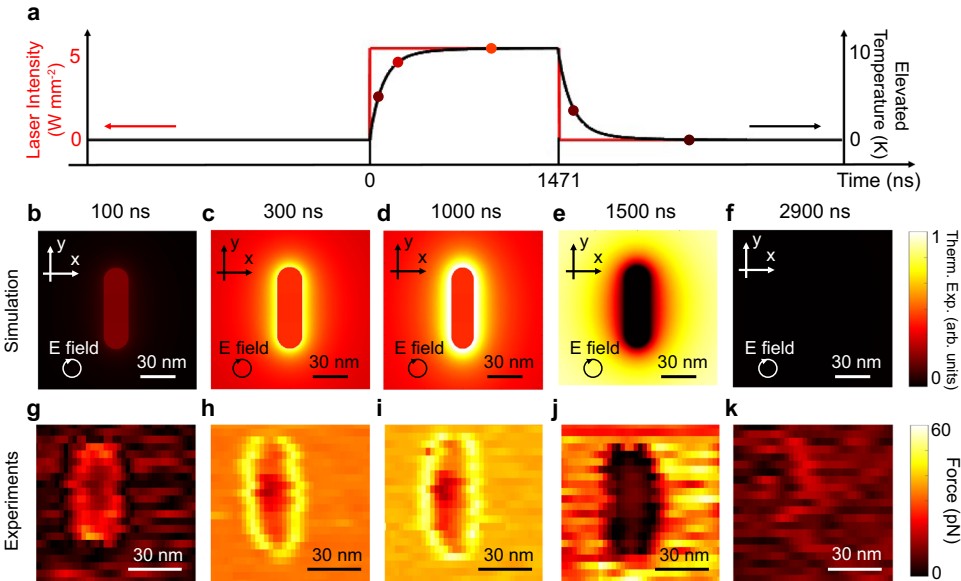

**Fig. 4 | Photothermal force maps show the temporal evolution of the nanoscale thermal profile. a** The average elevated temperature of the nanorod and the laser intensity as a function of time within one period. Simulated thermal expansion at the time frame of (**b**) 100 ns (with respect to the beginning of the optical pulse), (**c**) 300 ns, (**d**) 1000 ns, (**e**) 1500 ns, and (**f**) 2900 ns. **g–k** Measured photothermal force map dynamics at $f_{opt} - f_0$ of −1.5 kHz, and $f_d - f_0$ of 0.95 kHz, 0.75 kHz, 0.65 kHz, 0.25 kHz, and −0.15 kHz respectively. The free air amplitude is 100 mV, which corresponds to 15.9 nm. The measurement uses a setpoint of 70%.

distributions. Moreover, Dofn enables the spectroscopic analysis of emerging optical nanomaterials with unmatched spatiotemporal resolution, aiding in the characterization of nanophotonic devices, the design of nanoscale optical traps, and the innovation of thermal photonic systems.

## Methods

### Optical force measurement

The setup of the Dofn system is shown in Supplementary Fig. 1 with a detailed discussion in Supplementary Note 1. Briefly, we integrate an AFM (Asylum MFP-3D-BIO) with an inverted optical microscope (Olympus IX81). Circularly polarized light is focused on the sample with an objective lens (Olympus PlanApo 60x). The deflection of the cantilever containing the optical force information is demodulated by a lock-in amplifier (Signal Recovery 7280 DSP) with the reference frequency of $f_{opt}$. All measurements are proceeded in a transparent window with a diameter of 10 $\mu m$ to minimize the laser leakage to the photodetector inside the AFM head. A 2D frequency scan (both $f_d$ and $f_{opt}$) is measured to locate the optical modulation frequency $f_{opt}$ and the scanning range of the piezo's dithering frequency $f_d$ that maximize the amplitude. The sensitivity of the optical force measurement ranges from 0.5 pN to 10 pN depending on the difference between $f_{opt}$ and $f_d$. A smaller difference leads to a higher sideband leakage of the cantilever's deflection at $f_d$ to the lock-in frequency of $f_{opt}$, and therefore, reduces the sensitivity of the optical force measurement (Supplementary Note 9). The lock-in amplifier utilizes a reference frequency the same as the fundamental harmonic of the square wave, therefore, excluding the potential high-order harmonics. To measure the 2D optical force map of the nanorod, both amplitude and phase from the lock-in amplifier are recorded using a LabVIEW program continuously with a sampling frequency of 20 Hz and a scan frequency of 10 Hz. The 1D data is then mapped to a 2D map with the known scanning line number of 32. As shown in Supplementary Note 10, we assume a constant thermal drifting speed to correct the thermal drifting.

### Sample preparation

The nanorods were synthesized using a procedure reported in ref. 30. First, a solution of small gold seeds is prepared by mixing 9.75 mL of 0.1 M CTAB and 0.25 mL of 0.01 M HAuCl₄. 0.6 mL ice-cold 0.01 M NaBH₄ is added to the gold/CTAB solution. The solution is stirred for 10 min and aged for 1 h. Second, a growth solution is prepared by mixing 0.9 g of CTAB, 0.155 g of sodium oleate, and 25 mL of nanopure water at 40 °C. After cooling to room temperature, 2.4 mL of 4 mM AgNO₃, 25 mL of 1 mM HAuCl₄, and 0.21 mL concentrated HCl are added. Third, 125 μL of 0.064 M ascorbic acid is added to the growth solution. When the solution turns completely colorless, 20 μL of seed solution is added. The nanorods grow undisturbed for 16 h. Finally, the sample is rinsed 3 times with centrifugation at 1200 × g for 25 min, followed by diluting 10 mL of purified gold nanorods in 1 mM CTAB by adding 30 mL of nanopure water, then adding 1 mL of 60 mg/mL of 5000 MW PEG-Thiol in nanopure water. The solution is mixed gently overnight for 24 h. We centrifuge the solution at 900 × g for 25 min and disperse it into 10 mL of water. The nanorod sample is drop casted to a glass cover slip with the windows and PMMA fabricated on top (Supplementary Fig. 2). The sample is dried in vacuum at room temperature and washed with DI water.

## Data availability

Source Data file has been deposited in Figshare[31] under accession code https://doi.org/10.6084/m9.figshare.24175215. Source data are provided with this paper.

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

## Acknowledgements
The authors would like to thank Dr. Glennys Mensing, Dr. Mauro Sardela, and Dr. Kathy Walsh for their technical support in the cleanroom and with AFM. The authors also extend our gratitude to Hsuan-Kai Huang, Shengyan Liu, Yu Huang, and Wenning Fu for their technical assistance during certain stages of the experiments. The experiments were conducted in the Holonyak Micro and Nanotechnology Laboratory and the Material Research Laboratory Central Research Facilities at the University of Illinois Urbana-Champaign. Financial support for this work was provided by NIGMS R21GM139022 (to Y.Z.), the Innovative Science Accelerator Program (ISAC, www.isac-kuh.org) grant DK128851 (to Y.-S.C.), NSF CHE-2107793 (to C.J.M.), and the Hong, McCully, and Allen Fellowship (to H.W.).

## Author contributions
Y.Z. conceived the original idea and supervised the entire study. H.W., Y.-S.C., and Y.Z. designed and constructed the Dofn system. S.M.M. synthesized the gold nanorods under the direction of C.J.M. H.W. carried out the theoretical calculations, conducted the experiments, and analyzed the data. H.W., Y.-S.C., and Y.Z. contributed to the discussions and analyses of both theoretical and experimental results. All authors contributed to the writing of the manuscript.

## Competing interests
The authors declare no competing interests.
