## [Peer Review File · Nature Communications]

Visualizing Ultrafast Photothermal Dynamics with Decoupled Optical Force NanoscopyReviewer #1 (Remarks to the Author):

The manuscript presents AFM images of a plasmonic gold nanorod, whereby the rod is subjected to time-modulated laser illumination. Light absorption gives rise to periodic temperature variations of the rod and of its environment, leading to a modulation of the AFM signal provided by the cantilever. These periodic variations are isolated by a lock-in amplifier.

The design of the experiment is basically the one of ref. 8, with gold nanorods used as samples. The measurements, in particular the visualization of the plasmonic hot spots, appear of a good quality and are potentially interesting to a specialist audience. However, the main claim to novelty of the paper is the interpretation of these results, which advertises "visualizing ultrafast dynamics". I do not think the results and discussion of the paper support such a claim. I find the interpretation unclear and unconvincing, as explained below.

The discussion and interpretation of the results are unclear and difficult to follow. The authors postulate three forces (optical, photothermal, and photoacoustic). For me, only the optical force has a clear physical origin. The photothermal force is actually a displacement due to thermal expansion, re-interpreted as a force by introducing the spring constant of the cantilever. The origin of the 'photoacoustic' force is less clear (see detailed discussion in remark 3 below). No effort is made to estimate the relative magnitude of these forces and to prove that all of them are important.

My conclusion is that the experimental results are potentially interesting to a specialist readership, but that their interpretation and discussion do not go beyond those of Ref. 8. Therefore, the paper is incremental and unlikely to be of interest to the broad readership of Nat Comm. I recommend submission to a more specialized journal, after considerable improvement of the physical discussion and taking into account the following comments.

Detailed remarks

1. Single plasmonic particles have been observed by optical pump-probe microscopy on a true sub-picosecond time scale for more than 15 years. Therefore, the remark in the abstract that "transient photothermal effect at the nanoscale has not been observed [yet]" is not correct.

2. On page 3, the purported relation between scattered light and radiation pressure is not clear and should be explained. A simple relation exists for particles smaller than the wavelength, but it is a stretch to apply it to the system considered here.

3. The discussion on page 4 of three types of optical forces (photothermal, photoacoustic and radiation pressure) is very confusing, as it seems to place these three effects on the same fundamental level. I strongly disagree with the description presented, as discussed below:

i) "Radiation pressure" is understood here as the optical force, which includes gradient force and true radiation pressure effects. I agree that this is a true fundamental force between two objects in an electromagnetic field, arising from the balance of electromagnetic momentum. I am not sure the modeling in terms of optical potential is correct and would prefer a simulation including the balance of momentum transfer from the field to the tip.

ii) The "photothermal force" is not a real force in my opinion. It arises from thermal expansion, a fundamental process, but only appears as a force because of the change of position of the sample is translated into a force through the spring constant of the cantilever. This "force" would disappear if the sample position was kept constant.

iii) I had a hard time understanding what the authors call the "photoacoustic force". I first thought of the force at work in optical tweezers (see for example the Wiki link: [https://en.wikipedia.org/wiki/Acoustic_tweezers#:~:text=Acoustic tweezers \(or acoustical tweezers,can be called acoustical tweezers\)](https://en.wikipedia.org/wiki/Acoustic_tweezers#:~:text=Acoustic+tweezers,(or+acoustical+tweezers,can+be+called+acoustical+tweezers).)). However, this force would scale as the square of the pressure variation of the fluid and therefore should not change sign during the cycle as shown in Fig.1.

Rather than "photoacoustic force", I now see this force as a momentum exchange with the medium around the sample and tip (air in the case at hand). As the acoustic wavelength (micrometers) is much larger than the system, however, a conventional interpretation in terms of phonon momentum transfer is not very physical. A better description would be the hydrodynamic forces induced by the medium through air pressure and viscosity. Temperature-induced displacements of the sample lead to variations of the gap, which induce pressure changes (therefore a force on the tip area) and viscous drag forces. Both these forces should scale as the velocity of the substrate with respect to the tip, and therefore as the first derivative of the temperature. I do not see why a second derivative would be involved.

If the previous explanation is not correct in the authors' opinion, they should explain why and rewrite their manuscript in clearer terms.

4. The sentence on page 4 starting by Figure 1a can be misread as "the mechanical resonance frequency of the AFM probe is in the kHz regime", whereas it is the difference frequency between modulation and resonance which is kHz.

5. Page 5: the potential energy of the optical gradient force on a small particle scales as the product of polarizability and field squared (as correctly written in the SI). The force is a gradient of this quantity. Moreover, it is not clear whether this relation still holds for a much larger object, such as an AFM tip (see remark below about SI).

6. The expansion of the sample in a gradient of temperature is a complex elasticity problem (see sketch in ref. 8). In this paper, it has not been included in the COMSOL simulation, but has been approximated with equation (2). This is a very crude approximation, which is incorrect in my opinion.

7. The origin of equation (3) page 6 is completely unclear to me (see remark 3 iii above). It should be carefully argued and explained why the pressure and drag forces which should scale as the first derivative dT/dt are ignored.

8. Considering the coarseness of the model, and the non-exponential relaxation in the heat equation, accurate values for relaxation times (such as 282.9 ns) are not scientific.

9. Page 7, the decomposition of the force in three components on the basis of phases has too many unknowns if only one amplitude and one phase are given by the lock-in signal.

10. The title and conclusion suggest that the measurements are done with a time resolution of nanoseconds, which is definitely not the case. Variation of the lock-in frequency indirectly informs on the time-response of the different force components, but this analysis cannot replace a time-resolved investigation with pulses. This should be honestly acknowledged and explained.

11. Supplementary Information:

- I think the COMSOL simulations should include elastic deformation in addition to temperature.
- Why is the sample limited to 50 nm radius? The heat diffusion continues on larger scales, yielding relaxation over times that grow as the square of the length scale. The effect of previous illumination cannot be ignored.
- The modeling is done with a single light pulse, whereas the actual experiments are performed with a square wave. The accumulation of thermal energy in the sample is not properly accounted for.
- The expression for the gradient force is derived from that for a nanoparticle much smaller than the wavelength. There is not guarantee that it applies to the AFM tip with and that the force is independent of the tip position as assumed here. What is E_z ?
- Note 8: the time 32.7 ns is an uncertainty in phase, not a time resolution! The method cannot record a time trace on a time scale of nanoseconds.

Reviewer #2 (Remarks to the Author):

The authors report a method to map the photothermal forces with a spatial resolution at nanometer scale by using optical force nanoscopy. They show the phase responses of the photothermal force under temporal modulation of light. The manuscript reports a further interesting effect where the back-action of the photothermal effect can be used to obtain the dynamical photothermal process of a single gold nanorod in the nanosecond regime. The method developed in this manuscript is new and of broad interests to optics community.

The manuscript is written well with figures of high quality and good structure. The results are interesting, and experimental and simulated data support the conclusions. I would recommend publication of this work in Nature Communications. I suggest that the authors consider the following comments in their final submission:

1. The photothermal effect is usually considered as a specialized effect. It is worth to clearly address its suitability for publication in Nature Communications instead of a more focused journal in regards of the research interest to broad scientific community.
2. The example test mass in this manuscript is a gold nanorod. Can authors explain what those test masses are designed for? Why it is essential to map the photothermal force for those substrates? What are the possible applications?
3. The main experimental results are demonstrated in a single nanorod. Can authors comment on if this technique can be applied to a target of different materials and dimensions? Is there a limitation?
4. It is not very clear how the optical pulse is generated and if the pulse window has a rise-up and rise-down response that adds an error to the photothermal response time.

Reviewer #3 (Remarks to the Author):

This study aims to separate different photoinduced forces between an AFM tip and a plasmonic nanoparticle. They present a method to separate the thermal expansion contribution and the optical gradient (dipole-dipole). The origin of optically-induced tip-sample forces has been the subject of many recent studies and the clarification and separation of these contributions would be highly impactful. However, several items must be clarified and explained in this study for it to be publishable. I have outlined my concerns below.

The assignment of the real part of the optical force to the optical gradient force is not consistent with recent literature which show that the optical gradient force is much weaker than the induced change in van der Waals force due to thermal expansion (Jahng, et al., *Anal. Chem.* 2018, 90, 18, 11054–11061). This mechanism presented in Jahng, et al. also explains the absorptive lineshape of the force spectrum, whereas the spectrum of the optical gradient force should follow the real part of the sample dielectric function (i.e. dispersive spectral profile). The authors should explain why the spectrum they observe is absorptive and not dispersive as predicted previously. The authors should also clarify what optical force contribution is calculated and plotted in Fig. 2k.

Regarding Fig. 3, the authors claim that the resonance is shifted due to back-action but how do they rule out heating and softening of the cantilever with increasing DC laser power? This would have the same effect as the data shown: f_0 would decrease and the linewidth would broaden. The authors would need to measure the cantilever resonance curves for these power levels and rule out that this effect.

The authors' procedure to compensate drift in their images is not appropriate. They should present the raw, uncorrected data, or omit the data points lost due to drift. The method of moving data points from one side of the image to another cannot be used.

I don't understand how the authors measure the force at different time delays (Fig. 4). Detuning the laser repetition rate off of resonance would not probe the same time delay at each pulse, it would only change how effectively the force excites the tip resonance. On this note, the authors should state what excitation frequency and the tip tapping frequency was used for the measurement in Fig. 2.

How do the authors account for the shift in the tapping phase during imaging? Unless the laser is well synchronized with the motion of the tip, changes in sample viscoelastic properties could leak into the force phase channel, especially at edges in the topography. Did the authors also measure the AFM phase image?

Minor concerns:

The integral in Eq. 2 is unclear. What distance is the integral calculated over?

There are many grammatical errors and confusing phrases throughout the manuscript. I recommend a careful proof-reading before resubmission.

**Authors' Responses to the Reviewers' Comments on manuscript NCOMMS-22-36661-T
“Visualizing Ultrafast Photothermal Dynamics with Decoupled Optical Force Nanoscopy”**

We appreciate the reviewers for taking the time to review our manuscript and providing their constructive comments. We have carefully addressed all their questions and comments point-by-point and have made the necessary revisions to our manuscript. These revisions are highlighted in red within the manuscript and also listed in this response letter.

Reviewer #1:

Rev 1: The manuscript presents AFM images of a plasmonic gold nanorod, whereby the rod is subjected to time-modulated laser illumination. Light absorption gives rise to periodic temperature variations of the rod and of its environment, leading to a modulation of the AFM signal provided by the cantilever. These periodic variations are isolated by a lock-in amplifier.

The design of the experiment is basically the one of ref. 8, with gold nanorods used as samples. The measurements, in particular the visualization of the plasmonic hot spots, appear of a good quality and are potentially interesting to a specialist audience. However, the main claim to novelty of the paper is the interpretation of these results, which advertises “visualizing ultrafast dynamics”. I do not think the results and discussion of the paper support such a claim. I find the interpretation unclear and unconvincing, as explained below.

Authors: We thank the reviewer for recognizing the “good quality” and “potentially interesting” aspects of our research.

We respectively disagree with the reviewer that our experimental setup is somewhat similar to the one in Ref. 8, although we both used a modulated laser interacting with a sample and an engaged AFM probe. Our design principle and the technique for data extraction are very different from Ref. 8. In the following table, we made a side-by-side comparison between Ref. 8 and our approach to address the reviewer’s concerns related to Ref. 8.

Additionally, in this revised manuscript, we have included new experimental results and simulations to support our claim, in light of the reviewer’s comments. We hope our new results and revision will help to clarify our idea and make our concept easier to follow.

Table 1. Comparison of the methods

	Ref. 8	Our manuscript
Measured optical forces	Ref. 8 measured the force generated by light which is a mixture of the optical gradient force, photothermal force, and photoacoustic force . The paper brings up the multiple origins of the forces but did not measure them separately.	Our method decouples the optical gradient force, photothermal force, and photoacoustic force , and measures each force component. This is the first time these forces have been decoupled and measured separately. Furthermore, our measured photothermal force contains temporal information with a nanosecond resolution.
Modulation method	Ref. 8 used an off-resonance modulation. The laser is modulated with a relatively low frequency, ν_r , to drive the resonance	We used a near-resonance modulation. We modulated the laser near the resonance frequency and intentionally

Demodulation method	through the frequency-mixing effect with the cantilever's oscillation by the piezo's dithering at the frequency of ν_d.	chose a frequency slightly different from the piezo's dithering frequency to control the back-action of the photothermal expansion (Figure 3).
Tunability	Ref. 8 used a reference frequency of $n\nu_d + \nu_r$ to demodulate the signal through a lock-in amplifier. It is important to note that the phase information of the optical force is lost using this approach due to the difference between the two frequencies: ν_r and $n\nu_d + \nu_r$.	We used the laser modulation frequency as the reference frequency of a lock-in amplifier. The phase information from the optical force is preserved, which is critical to decouple the different optical force components (optical gradient force, photothermal force, and photoacoustic force).
Samples	The wavelength of the laser can be changed to measure spectral information in mid-IR. Ref. 8 used a fixed modulation frequency, as the modulation frequency in their setup may not influence the tip-sample interaction. No temporal information was presented.	We scanned the wavelength of the laser to obtain spectral information in the visible regime. Importantly, we changed the piezo's dithering frequency to control the back-action of the photothermal expansion. Therefore, we can acquire not only the spectral information but also the temporal information of the photothermal force.
	Ref. 8 measured a polymer thin film, where the optical forces are uniform across the sample and high spatial resolution is not required. The origin of the optical force is mostly from the photothermal expansion.	We measured a plasmonic nanoantenna because this sample involves relatively more complicated optical force components to demonstrate our technique. The measured three types of optical forces are in similar magnitudes, but the distribution is heterogeneous across the sample. This nanoscale sample requires high spatial resolution. Our measurement is not limited to plasmonic samples, but a wide range of nanomaterials, for example, 2D materials, nanophotonic devices, light-sensitive nanoparticles, light-absorbing molecules, etc.

Rev 1: The discussion and interpretation of the results are unclear and difficult to follow. The authors postulate three forces (optical, photothermal, and photoacoustic). For me, only the optical force has a clear physical origin. The photothermal force is actually a displacement due to thermal expansion, re-interpreted as a force by introducing the spring constant of the cantilever. The origin of the

'photoacoustic' force is less clear (see detailed discussion in remark 3 below). No effort is made to estimate the relative magnitude of these forces and to prove that all of them are important.

Authors: we thank the reviewer for pointing out his/her confusion about the physical origins of the optical forces, especially the photothermal force and photoacoustic force. The photothermal force is caused by thermal expansion due to optical absorption that is proportional to the overall thermal expansion of the nanorod and the tip of the AFM probe. The photoacoustic force is caused by the pressure wave that originates from the rapid thermal expansion and contraction of the sample. The photoacoustic wave interacts with the entire AFM probe.

We have briefly explained the physical origins of these forces in the main text, page 8 of the manuscript, "*The optical gradient force is associated with the electrical polarization of the AFM probes as well as the electrical field intensity in the z-direction at the AFM tip location*" and "*The photothermal force is proportional to the overall thermal expansion of the nanorod and the AFM probe at the nanoparticle-tip interface*". In addition to our explanation, the physical origins of these forces have been recognized in Refs. 8, 24, 25, 26, and SI Ref. 8.

As the reviewer mentioned that the photothermal expansion causes the deflection of the cantilever, which will be perceived by the cantilever as a force in the measurement. In our setup, the photothermal expansion is associated with the optical modulation frequency of f_{opt} , and the cantilever piezo is modulated at the dithering frequency of f_d . Therefore, when thermal expansion occurs, the gap between the tip and sample varies, causing a tip-sample interaction, which causes a deflection to the cantilever. Since the cantilever deflection reflects the overall forces, the photothermal force and the rest of the light-induced forces will be measured together. Furthermore, the photothermal effect could generate a thermal gradient force. The thermal gradient force has been used to trap particles and molecules in solution (Chen, J., Cong, H., Loo, FC. et al. Thermal gradient induced tweezers for the manipulation of particles and cells. *Sci Rep* 6, 35814 (2016). <https://doi.org/10.1038/srep35814>). Although the sample that we measured is in the air and the thermal gradient force is minor, our method can be used to extract the forces associated with the photothermal effect of the specimen, which can also be used to estimate the photothermal force in these trapping schemes.

To further explain the photoacoustic force, we conducted additional simulations (Supplementary Note 4 Photoacoustic simulation, and Supplementary Fig. S8 as quoted below). Our simulation shows that the pressure wave interacts with the cantilever and generates a force magnitude similar to our measurements. We also revised the sentence on page 6 of the main text to make it clearer. The revised sentence reads, "*As a result, the photoacoustic pressure will exert on the entire probe instead of a localized spot near the tip, resulting in a non-localized photoacoustic force (Supplementary Note 4), which can be treated as a uniform background .*" Our simulations in supplementary figures S7c confirmed this assumption.

"Supplementary Note 4: Photoacoustic simulation

Figure S7c shows the photoacoustic force is a uniform background force. In the measurement, the total photoacoustic background force is contributed by an ensemble of nanorods in the illumination area. To further understand the order of magnitude of the photoacoustic force and its temporal profile, we simulate the photoacoustic pressure given by equation (7) through the COMSOL PDE module. We use a uniform heat source that approximately matches the absorbed power of nanorods in the illumination area, where the incident light is modulated with a square wave with a peak intensity of 5 MW/m^2 , as shown in Fig. S8a (dashed curve, right y-axis). The AFM probe is modeled the same as the one used in the experiment (OPUS 4XC-NN, standard AC mode cantilever). We calculate the photoacoustic force by integrating the photoacoustic pressure over the surface area of the AFM probe:

$$\mathbf{F}_{PA} = \iint_{probe} P ds, \quad (7)$$

where P is the photoacoustic pressure on the AFM probe surface. As shown in Fig. S8a, the simulated photoacoustic force shows two opposite peaks following the rising and falling edges of the square wave modulation as expected. The pressure distribution at the peak position shows that the force is mainly probed by the sample-side surface of the cantilever (Fig. S8b). The distortion of the waveform is attributed to the acoustic wave scattering by the AFM probe. The simulated photoacoustic force shows odd symmetry with respect to the center of the square wave, as illustrated by the phase relationship of the three types of optical forces in Figure 1. The simulated PA force shows a peak intensity of 1.39 nN and a full width at half maximum of 79 ns. The average force in a period of around 5.9 μ s is 37.2 pN. This estimation is close to our measured background force in Figure 2b of around 50 pF. The relatively lower value of the simulation may be attributed to that we ignored the photoacoustic force from the absorption of the AFM probe and the PMMA film, which has a relatively high thermal expansion coefficient.

Supplementary Fig. S8 | Multiphysics simulation of the photoacoustic force. **a**, Simulated photoacoustic force. The solid curve in the left plot represents the simulated photoacoustic force, F_{PA} , and the dashed curve represents the surface power intensity. **b**, Simulated photoacoustic pressure distribution. ① and ② correspond to the peak positions marked in **a**.

The relative amplitude of these optical forces can be seen in Figure 2. As shown in Figures 2d and 2e, the photothermal force is in a similar amplitude to the optical gradient force. The amplitude of the photoacoustic force, as the background force in Figure 2b, is estimated to be ~ 50 pN. To clarify the force magnitude, we included additional details “The three optical forces have similar magnitudes. The photothermal force has a peak intensity of 35.7 pN (Figure 2d), and the optical gradient force has a peak intensity of 30.6 pN (Figure 2e). The photoacoustic force, as the background force in Figure 2b, has an averaged intensity around 50 pN.” on page 8 of the revised manuscript.

Rev 1: My conclusion is that the experimental results are potentially interesting to a specialist readership, but that their interpretation and discussion do not go beyond those of Ref. 8. Therefore, the paper is incremental and unlikely to be of interest to the broad readership of Nat Comm. I recommend submission

to a more specialized journal, after considerable improvement of the physical discussion and taking into account the following comments.

Authors: we have listed a detailed comparison between our work and Ref. 8 in our answers to the reviewer's first comment. Additionally, we have listed all the figures from Ref. 8 below. The main goal of Ref. 8 was to differentiate optical dipole force from thermal expansion-induced impulsive force using photoinduced force microscopy (Figure 1). They showed the measured optical force curve (i.e., the overall force) (Figure 2a), the theoretical optical gradient curve (Figure 2b), and a force map (the overall force) of a PMMA polymer film. They further calculated the photothermal and optical gradient as a function of polymer film thickness and overlay the calculated force curves with measure forces at various film thicknesses (Figure 3). This measured force is a mixed force from the three components. Finally, they compared the overall optical force curve in two different thicknesses of the polymer film (Figure 4). The authors also presented a table to compare the three main force components, i.e., the thermal expansion, optical gradient force, and photoacoustic force, which we measured separately in our study. The table listed in Ref. 8, however, is an illustration for a conceptual discussion.

We want to reiterate that the concept of our approach is very different from the existing literature. There are two important aspects of our technique. First, we use a square wave to modulate the laser in order to separate the optical forces in the complex domain. Second, we modulate the laser at a frequency slightly different from the piezo's dithering frequency. The first aspect allows us to decouple the different force components and the second aspect gives rise to the tuning capability to study different time frames in the nanosecond regime and at the same time, maintain the nanometer spatial resolution, as we have compared in the table above.

Furthermore, to highlight the main contributions of our approach, we have included additional simulations and experimental results to address the reviewer's comments and strengthen our claim. In summary, we have employed a different modulation technique to delineate optical forces from different physical origins and visualize ultrafast photothermal dynamic distributions across a single nanoparticle. This technique has not been reported in the literature.

Figure 1 of Ref 8. (a) Photoinduced force experiment: mid-infrared light pulsed at repetition rate ν_r illuminates the tip-sample gap with possible optical dipole force F_{opt} and thermal expansion-induced impulsive force $F_{th.exp}$ indicated and simultaneous s-SNOM mode collecting tip-scattered near-field. (b) Lock-in amplifier receives tip dither frequency ν_d , tip motion $h(t)$, and s-SNOM near-field signal NF in demodulating at harmonics of the tip motion $n\nu_d$. Fourier components of (c) s-SNOM signal and (d) cantilever dynamics with sidebands at frequencies $\nu_d \pm \nu_r$ due to mixing of cantilever motion and the photoinduced forces.

Figure 2 of Ref. 8. (a) Measured force spectrum of carbonyl resonance of PMMA (blue) with corresponding Lorentzian fit (dashed) matches the imaginary part of the index of refraction $Im(\tilde{n}) = \kappa$ (green). (22) (b) Theoretical optical gradient force spectrum calculated using eq 1 and the real part of index of refraction $Re(\tilde{n})$ both show dispersive behavior. (c) AFM topography, (d) s-SNOM amplitude, (e) and force signal of a PS-PMMA block copolymer sample with laser tuned to the carbonyl resonance at $\nu_{C=O} = 1735 \text{ cm}^{-1}$. The force signal shows high contrast with strong signal on PMMA with $\sim 25 \text{ nm}$ spatial resolution and correlated with the s-SNOM amplitude (f).

Figure 3 of Ref. 8. Photoinduced force dependence on PMMA thickness (blue symbols). The relative thermal expansion calculated from eq 3 (blue line), and optical gradient force calculated for the PMMA/Si layered system (green, solid) and for a free-standing PMMA layer (green, dashed) are shown for comparison.

Figure 4 of Ref. 8. a) Photoinduced force approach curves on a 60 nm thick PMMA film (black), 10 nm thick PMMA film (blue), and Si (red) at the carbonyl resonance ($\bar{\nu} = 1735 \text{ cm}^{-1}$). For each sample, a prominent increase in signal occurs only at the point of tip-sample contact ($z = 0 \text{ nm}$). Only for the 60 nm film, a long-range force described by an exponential decay (magenta) with decay length $l = 25 \text{ nm}$ is observed. Predictions from the point-dipole model for the optical gradient force are shown in green. (b) Approach curve measurement over a PTFE surface, with laser tuned to resonance with the C—F symmetric stretch at $\bar{\nu} = 1152 \text{ cm}^{-1}$. A weak attractive force is observed before the onset of the repulsive thermal expansion.

Table 1 Key Characteristics of Photo-Induced Forces along with the Predicted Characteristics of the Force Due to Direct Thermal Expansion, the Optical Gradient Force, and the Photoacoustic Force

	Thermal expansion	Optical gradient force	Photoacoustic force
Force spectrum			
Distance dependence			
Sample thickness			

Detailed remarks (Reviewer 1):

1. Single plasmonic particles have been observed by optical pump-probe microscopy on a true sub-picosecond time scale for more than 15 years. Therefore, the remark in the abstract that “transient photothermal effect at the nanoscale has not been observed [yet]” is not correct.

Authors: yes, single plasmonic particles have been observed for many years. However, observing the single plasmonic particles is very different from observing the nanoscopic photothermal map around a single nanoparticle. We have cited the abovementioned pump and probe microscopy in our manuscript as Ref. 39. Pump-probe microscopy characterizes the information directly related to the nanoparticle's electromagnetic far-field radiation. Although the photothermal effect can be inferred from optical absorption, the image itself is typically diffraction limited. For example, as shown in Figure 5 of Ref. 39, the single nanoparticle is observed as a diffraction-limited point spread function (please see the quoted image below). In other words, pump-probe microscopy itself does not provide sufficient resolution to visualize the distribution of the photothermal effect at the nanoscale as we reported in this manuscript.

Figure 5 of Ref 39. WPS imaging of etched pattern in PMMA film and microparticles. (A) Reflection image of the pattern, where the etched-off parts showed higher reflectivity. (B) WPS image of the same area. (C) First derivative of the intensity profile along the line is shown in (B) as squares. a.u., arbitrary units. Gaussian fitting (red line) showed an FWHM of 0.51 μm . (D) Reflection image of 1 μm of PMMA particles. (E) WPS image of the same area with the pump at 1728 cm^{-1} . (F) Off-resonance image showed no contrast. Scale bars, 10 μm .

2. On page 3, the purported relation between scattered light and radiation pressure is not clear and should be explained. A simple relation exists for particles smaller than the wavelength, but it is a stretch to apply it to the system considered here.

Authors: we thank the reviewer for raising a question about the scattered light and radiation pressure. Through further numerical simulations, we show that although the AFM probe is significantly larger than the wavelength, the interacting region is confined at the tip of the AFM probe, which is significantly smaller than the wavelength. The radiation pressure does not change regardless of the engagement. Therefore, it will be removed as a background force at the beginning. We have included the additional simulation results and description in the supplementary materials and added a short description on page 5 of the main text.

The additional sentences in the main text read, “Although the AFM probe is significantly larger than the wavelength, the interacting region is limited to the tip of the AFM probe as shown in Fig, S3 (Supplementary Note 2). Therefore, the relationship for small particles, such as described in Ref. 53, still holds true. The radiation pressure is independent of the engagement condition. It is removed at the beginning as a constant background force.”

The additional Supplementary note and figure are as follows:

“Supplementary Note 2. Optical gradient force simulation

In the main text equation (1), we model the AFM tip as a small particle for optical gradient force with a dipole approximation. One may argue that this is not generally true, because the size of the AFM probe is significantly larger than the wavelength. However, such an approximation remains valid when only the tip of the AFM probe is involved in the near-field interaction. To validate this approximation, we proceed with a full wave numerical simulation using the COMSOL Electromagnetics module. In the simulation, we used an AFM probe with a height of 500 nm. The simulated polarization density of the AFM probe and the nanorod is shown in Fig. S3. The proportion of the AFM probe's non-zero polarization density is less than 25 nm, as shown in Fig. S3. Consequently, if the AFM probe is modeled as a dipole, the error should not be substantial.

Supplementary Fig. S3| Simulated polarization density of the AFM probe and the nanorod. The AFM probe is made of silicon and has a tip radius of 5 nm, a height of 500 nm, and an aspect ratio (height/diameter) of 1.75. The tip of the AFM probe is kept 1 nm away from the surface of the nanorod.

We further simulate the 1D distribution of the optical force with the scanning line marked in Fig. S4a. We simulate the optical gradient force by the surface integral of the Maxwell stress tensor on the AFM probe

$$\mathbf{F} = \oint_{probe} \boldsymbol{\sigma} ds, \quad (1)$$

where $\boldsymbol{\sigma}$ is the Maxwell stress tensor on the AFM probe,

$$\boldsymbol{\sigma} = \left[\epsilon_0 \left(\mathbf{E} \otimes \mathbf{E} - \frac{1}{2} \mathbf{E}^2 \mathbf{I} \right) + \frac{1}{\mu_0} \left(\mathbf{H} \otimes \mathbf{H} - \frac{1}{2} \mathbf{B}^2 \mathbf{I} \right) \right],$$

where \mathbf{E} and \mathbf{H} are the electrical and magnetic fields, \otimes represents dyadic product, and \mathbf{I} represents a 3-by-3 identity matrix. The simulated force distribution shown in Fig. S4b fits well with the measured and theoretical distribution shown in Figure 2f and 2i.

The dominant effect in generating the optical gradient force differs for dielectric samples¹ and plasmonic samples. For the dielectric samples, the force is mainly given by the lightning rod effect of the AFM tip apex and its induced-dipole moments on the substrate that is related to the dielectric properties of the substrate. On the other hand, for plasmonic samples, the optical force is mainly given by the localized electric field strength induced by the sample due to the plasmonic resonance.

Supplementary Fig. S4| Simulation of the optical gradient force. a, Simulation model. The gold nanorod is placed on top of a substrate composed of a 200 nm PMMA layer and a 500 nm SiO₂ layer. The AFM tip is kept 1 nm away from the sample surface. The scanning line is marked as the dashed line. b, Normalized force distribution in the 1D scanning line marked in a.

3. The discussion on page 4 of three types of optical forces (photothermal, photoacoustic and radiation pressure) is very confusing, as it seems to place these three effects on the same fundamental level. I strongly disagree with the description presented, as discussed below:

i) “Radiation pressure” is understood here as the optical force, which includes gradient force and true radiation pressure effects. I agree that this is a true fundamental force between two objects in an electromagnetic field, arising from the balance of electromagnetic momentum. I am not sure the modeling in terms of optical potential is correct and would prefer a simulation including the balance of momentum transfer from the field to the tip.

Authors: we thank the reviewer for suggesting a full-wave numerical simulation of the optical gradient force. We have included it in Supplementary Information Note 2. As shown in Supplementary Fig. S3 and Fig. S4, only the tip of the AFM probe is involved in the polarization, the result fits with our theoretical prediction with the dipole approximation (Eq. (1)).

ii) The “photothermal force” is not a real force in my opinion. It arises from thermal expansion, a fundamental process, but only appears as a force because of the change of position of the sample is translated into a force through the spring constant of the cantilever. This “force” would disappear if the sample position was kept constant.

Authors: we respectfully disagree that the photothermal force is not a real force. As shown in our previous response, the laser is modulated at a different frequency compared to the cantilever’s dithering. The feedback loop of the AFM system, using the deflection at f_d as the input, will not influence the separation by the thermal deflection at f_{opt} . Although the AFM does not operate at the contact mode, the

thermal expansion will result in a force through the tip-sample interaction. In addition, the photothermal force has been well documented in the field of force microscopy, please refer to the following references.

1. Aaron Katzenmeyer, Vladimir Aksyuk, and Andrea Centrone, Nanoscale Infrared Spectroscopy: Improving the Spectral Range of the Photothermal Induced Resonance Technique *Anal. Chem.* 2013, 85, 4, 1972–1979
2. Alexandre Dazzi, and Craig B. Prater, AFM-IR: Technology and Applications in Nanoscale Infrared Spectroscopy and Chemical Imaging, *Chem. Rev.* 2017, 117, 7, 5146–5173

iii) I had a hard time understanding what the authors call the “photoacoustic force”. I first thought of the force at work in optical tweezers (see for example the Wiki link:

[https://en.wikipedia.org/wiki/Acoustic_tweezers#:~:text=Acoustic tweezers \(or acoustical tweezers, can be called acoustical tweezers\).](https://en.wikipedia.org/wiki/Acoustic_tweezers#:~:text=Acoustic+tweezers+(or+acoustical+tweezers),can+be+called+acoustical+tweezers).) However, this force would scale as the square of the pressure variation of the fluid and therefore should not change sign during the cycle as shown in Fig.1.

Rather than “photoacoustic force”, I now see this force as a momentum exchange with the medium around the sample and tip (air in the case at hand). As the acoustic wavelength (micrometers) is much larger than the system, however, a conventional interpretation in terms of phonon momentum transfer is not very physical. A better description would be the hydrodynamic forces induced by the medium through air pressure and viscosity. Temperature-induced displacements of the sample lead to variations of the gap, which induce pressure changes (therefore a force on the tip area) and viscous drag forces. Both these forces should scale as the velocity of the substrate with respect to the tip, and therefore as the first derivative of the temperature. I do not see why a second derivative would be involved.

If the previous explanation is not correct in the authors’ opinion, they should explain why and rewrite their manuscript in clearer terms.

Authors: we thank the reviewer for sharing his/her insight on his/her postulation of the photoacoustic force’s possible physical origins. Because our measurement is in the air, the force due to the viscosity of the media is negligible. The photoacoustic force originates from the pressure generated from the sample surface due to photothermal heating and cooling. This force is different from the acoustic tweezers as described by the reviewer. The photoacoustic pressure is proportional to the second derivative of the temperature, Equation (12.11) on page 287 of Wang, L. V., & Wu, H. I (2012). *Biomedical optics: principles and imaging*. John Wiley & Sons. We have updated the citation for the sentence “*The acoustic pressure is proportional to the second time derivative of the photothermal expansion.*” with the one listed above.

We quote the key discussion from the book for the reviewer’s reference:

The volume expansion, dV / V , can be expressed as Eq. (R1)

according to Hooke’s law.

$$\frac{dV}{V} = -\kappa p + \beta T, \tag{R1}$$

where κ denotes the isothermal compressibility, $\kappa = \frac{C_p}{\rho v_s^2 C_v}$.

The differential volume can be expressed in terms of the medium displacement $\vec{\xi}$,

$$\frac{dV}{V} = \nabla \cdot \vec{\xi}. \quad (\text{R2})$$

The displacement vector is related to the pressure through the inviscid flow equation:

$$\rho \frac{\partial^2 \vec{\xi}}{\partial t^2} = -\nabla p. \quad (\text{R3})$$

Taking the divergence of Eq. (R3), we get

$$\rho \frac{\partial^2}{\partial t^2} [\nabla \cdot \vec{\xi}] = -\nabla^2 p \quad (\text{R4})$$

Substituting Eq. (R1) and Eq. (R2) into Eq. (R4), we get the photoacoustic equation (as described in Supplementary Note 4, Eq. (3))

$$\left(\nabla^2 - \frac{1}{v_s^2} \frac{\partial}{\partial t^2} \right) p(\mathbf{r}, t) = -\frac{\beta}{\kappa v_s^2} \frac{\partial^2 T(\mathbf{r}, t)}{\partial t^2}, \quad (\text{R5})$$

As shown in equation (R5), the photoacoustic pressure $p(r, t)$ originates from the second time derivative of the temperature $\frac{\partial^2 T(\mathbf{r}, t)}{\partial t^2}$, and thus it is crucial to use a modulated laser in *pulses*, not a modulated laser with a sinusoidal function, nor a CW laser to generate the photoacoustic force. Furthermore, the square wave modulation creates two opposite photoacoustic pulses following the rise and fall edges. The difference in symmetry of the pulse's temporal profile enables us to decouple it from the other optical forces.

It is not correct that the acoustic wavelength is much larger than the system. The wavelength of the acoustic wave is not in mm, we made a typo in the original manuscript, and we apologize for the confusion. Figure 1c shows the numerical simulation of the photoacoustic pressure. As the photoacoustic pressure is generated by the second derivative of the temperature, the photoacoustic pulse width, as shown in Figure 1c, is around 471 ns. The acoustic pulse length in air is therefore approximately 160 μm . It is on the same length scale as the cantilever's size of around 175 μm but much larger than the tip radius of around 20 nm. It is important to note that the photoacoustic force is not given by the localized tip-sample interaction but is generated by many gold nanorods that are under illumination and measured by the whole cantilever.

To avoid such confusion, we have revised the description on page 7 of the main text to “*As the photoacoustic pressure is generated by the second time derivative of the temperature, the photoacoustic pulse width, as shown in Figure 1c, is around 471.0 ns. The pulse length in air is therefore around 160.1 μm . It is on the same length scale as the cantilever's size of around 175 μm but much larger than the tip size of around 20 nm.*” and included additional simulations to *Supplementary Note 4: Photoacoustic simulation*, in the revised SI (as shown in the first question).

4. The sentence on page 4 starting by Figure 1a can be misread as “the mechanical resonance frequency

of the AFM probe is in the kHz regime”, whereas it is the difference frequency between modulation and resonance which is kHz.

Authors: We thank the reviewer for pointing out this confusion. The cantilever resonance and modulation frequencies are both around 170 kHz. We have revised the wording from “the kilohertz regime” to “*the hundred-kilohertz regime*”.

5. Page 5: the potential energy of the optical gradient force on small particle scales as the product of polarizability and field squared (as correctly written in the SI). The force is a gradient of this quantity. Moreover, it is not clear whether this relation still holds for a much larger object, such as an AFM tip (see remark below about SI).

Authors: We thank the reviewer for raising the question about the approximation in the optical gradient force. To address this question, we have conducted a full-wave numerical simulation to calculate the optical gradient force. The new results are included in Supplementary Note 2 as we addressed in answering the reviewer’s second question. Briefly, although the size of the AFM probe is significantly larger than the wavelength, our full wave numerical simulation (COMSOL Electromagnetics module) shows that the AFM probe’s non-zero polarization density covers a volume with a diameter smaller than 25 nm, as shown in the Supplementary Fig. S3 (also shown above in the answers to question NO. 2). Consequently, if the AFM probe is modeled as a dipole, the error should not be substantial.

6. The expansion of the sample in a gradient of temperature is a complex elasticity problem (see sketch in ref. 8). In this paper, it has not been included in the COMSOL simulation, but has been approximated with equation (2). This is a very crude approximation, which is incorrect in my opinion.

Authors: We agree with the reviewer that the thermal expansion is a complex problem and thank the reviewer for suggesting additional simulations. We have included additional numerical simulations to validate our model suggested by equation (2). We compare our theoretical calculation from equation (2) with the simulated deflection of the cantilever using COMSOL Solid Mechanics Module. Our simulation matches well with the results using equation (2) and confirms that the thermal expansion is small compared to the cantilever’s size, and therefore, thermal expansion can be approximately modeled by equation (2).

We have included these additional simulation results in Supplementary Note 3 and also attached them below. Because the dimensions of a single nanorod and a cantilever mismatch dramatically, where the cantilever is hundreds of times larger than a nanorod, and thus the computational burden is significant due to this mismatch in sizes and mesh densities. To reduce the computational burden, we increase the size of the nanorod by 10 times while keeping the elevated temperature of the nanorod the same as Figure 4a. This temperature is calculated based on the *actual* nanometer size of the nanorod. Note that this is valid because the nanometer size of the nanorod only affects spatial confinement besides the temperature. As we have kept the temperature the same, increasing the size is an appropriate approximation. The simulated displacement of the cantilever from the thermal expansion shows a slight difference from the calculated one given by equation (2) by less than 20%. This difference mainly comes from the anisotropic expansion of the tip and the sample along the vertical direction (z) and would be smaller for the actual size because the size of the nanorod and the heated region is smaller. Based on these calculations, we believe equation (2) provides an adequate physical understanding of the photothermal force, and therefore, we have kept this equation in our manuscript.

“Supplementary Note 3. Photothermal simulation

The photothermal force is generated by the thermal expansion around the nanorod. In equation (2) of the main text, we simplify such an expansion as an isotropic displacement along the probing direction (dashed line along the z-direction as shown in Fig S6a). Due to the irregular shape of the sample and the AFM tip, such a simplification may yield an error compared to the simulated photothermal expansion-induced displacement of the cantilever. As shown in Fig. S6a, we simulate the thermal expansion induced by the heated nanorod with an elevated temperature of 10.9 K (close to the one shown in Figure 4a in the main text). Because the size of the nanorod (90 nm in length and 30 nm in width) is over three orders of magnitudes smaller than the AFM probe (170 μm in length and 40 μm in width), meshing the entire domain to achieve an accurate calculation will cause a significant computational burden. To reduce the computational burden and the meshing difficulty, we increase the size of the nanorod by 10 times but keep the temperature elevation the same as our previous calculation based on the actual nanorod dimension. We simulate the displacement of the cantilever using COMSOL Solid Mechanics Module (Fig. S6b). The surrounding 4 surfaces of the substrate and the base of the cantilever are set as fixed boundaries. The temperature distribution along the probing direction is shown in Fig. S6c.

We calculate the displacement with equation (2) and compare it with the simulated displacement (Fig. S6d). The simulated displacement of the cantilever from the thermal expansion shows a difference from the calculated one given by equation (2) by less than 20%. The difference mainly comes from the non-isotropic expansion of the tip and the sample along the z-direction and would be smaller for the actual size because the heat will be spatially more confined. The spring constant of the cantilever was measured to be 8.6 N/m. By multiplying the spring constant and diving by the scale factor of 10, we get an approximate photothermal force of 70.8 pN, which roughly fits our measurement. The slightly bigger value from the theoretical calculation may be attributed to the less confined temperature distribution because we used a larger rod in the simulation compared to the experiment.

Supplementary Fig S6 | Multiphysics simulation of the photothermal expansion. **a**, Simulated elevated temperature distribution. The elevated temperature of the nanorod is set as 10.9 K based on the simulation in Fig. S5. We increase the nanorod by 10 times to a length of 0.9 μm , and a width of 0.3 μm , and simulate the substrate with a 50- μm -thick glass layer. **b**, Simulated displacement distribution. The probing direction is marked as the dashed line. **c**, Temperature distribution along the probing direction. The subplot shows the position of the substrate ①, nanorod ②, and AFM tip ③. **d**, Theoretical and simulated thermal expansion along the probing line. The theoretical curve is calculated by equation (2).

7. The origin of equation (3) page 6 is completely unclear to me (see remark 3 iii above). It should be carefully argued and explained why the pressure and drag forces which should scale as the first derivative dT/dt are ignored.

Authors: As discussed in 3 iii, for a pulsed irradiation, the temperature rise in nanostructure and surrounding medium is calculated by the heat diffusion equation. The temperature rises of the nanostructure and its environment is used in structural mechanics model of linear thermal expansion to evaluate stress and strain tensor through the Hookes law. The stress tensor is related to the second derivative of the structural displacement with respect to time (Prost A et al. Photoacoustic generation by a gold nanosphere: From linear to nonlinear thermoelastics in the long-pulse illumination regime. *Physical Review B*. (2015) <https://doi.org/10.1103/PhysRevB.92.115450>). Therefore, the photoacoustic pressure generated by thermal expansion is proportional to the second derivative of temperature. The detail derivation of the equation is on page 287 of Wang, L. V., & Wu, H. I. (2012). *Biomedical optics: principles and imaging*. John Wiley & Sons; and the brief summary is shown in the answer to 3 iii.

The viscous force induced by the photothermal expansion is indeed proportional to the first derivative of the elevated temperature, dT/dt . However, as the measurement is performed in the air which has a relatively low viscosity, the viscous force is small compared to the other forces. Therefore, we ignored the viscous force.

8. Considering the coarseness of the model, and the non-exponential relaxation in the heat equation, accurate values for relaxation times (such as 282.9 ns) are not scientific.

Authors: We respectively disagree that our calculated relaxation times are not scientific and the non-exponential relaxation. We have listed our reasonings below.

To simulate the photothermal heating, we used the temporal width of the excitation laser instead of an impulse. The dimension of the nanorod is in nanometers and gold has a high thermal conductivity, therefore, heat generated within the nanorod will not be confined throughout the duration of laser pulses. It means that depending on the environment and the laser pulse width, heat often leaks out during the thermal heat deposition, thus, the thermal profile of gold nanorod and surrounding is time dependent (Y.-S. Chen et al, Environment-Dependent Generation of Photoacoustic Waves from Plasmonic Nanoparticles, *Small* (2011), <https://doi.org/10.1002/sml.201101140>). It is worth noting that our definition of thermal relaxation time is slightly different from the conventional thermal relaxation time in Ref. 6, which is usually defined as the time for the sample to decay to e^{-1} of the peak temperature when the sample is heated by an impulse, where all the heat is absorbed before leaking out. Instead, we define the thermal decay time as the time when the system temperature decays to 10% of the peak temperature. In this way, our definition of the thermal decay time will mimic the scenario measured during the experiment.

To clarify the definition of the thermal decay time, we revised our manuscript and included an additional discussion in Supplementary Note 3, which reads “*Please note that we define the thermal decay time as the time for the elevated temperature of the nanorod to decay to 10% of its peak elevated temperature with the square wave modulation in Fig. S5a. Due to the absorption waveform of the nanorod is not of a short impulse, the thermal decay time here is different from the conventional definition* ⁶.”

⁶ Yadav, R. K. (2009). Definitions in laser technology. *Journal of cutaneous and aesthetic surgery*, 2(1), 45-46.

The relaxation is indeed exponential. Due to the linearity of the system, the thermal relaxation process will always be exponential regardless of the heating time. As shown in Fig. R1, we compare the simulated temperature evolution of the nanorod with the theory, which matches well with each other. In theory, the temperature evolution follows $T = (1 - 10^{-t/\tau})T_{\max}$ during the heating phase and $T = 10^{-t/\tau}T_{\max}$ during the cooling phase, where τ is the thermal decay time of 282.9 ns and T_{\max} is the maximum elevated temperature of 10.9 K. We included our simulated relaxation time in Fig. 1c. We also show Fig. R1 below as evidence that the heating and cooling processes take a significant portion in a modulation period.

Fig. R1| Simulated temperature compared to the exponential relaxation process.

9. Page 7, the decomposition of the force in three components on the basis of phases has too many unknowns if only one amplitude and one phase are given by the lock-in signal.

Authors: We agree with the reviewer that there are more unknowns than the equations. In our approach, we utilize two known physical knowledge to facilitate the decoupling.

First, the photoacoustic pressure is relatively uniform across the scanning area of several hundred nanometers. As we shown in supplementary note 4, the photoacoustic pressure interacts with the entire cantilever, because the photoacoustic pressure wave travels in micrometer range during each modulation period, which results in a nearly uniform force measured by the AFM probe. In this way, we can decouple the photoacoustic force from the measurement. To further confirm our approach, we simulate the photoacoustic force measured by the AFM probe (Fig. S7 and Fig. S8). The result shows that the photoacoustic force is indeed approximately uniform across the scanning area. Therefore, the photoacoustic force can be removed from the total optical force as the background force.

Second, the photothermal force is dominant over the optical gradient force at the center of the nanorod, because the electric field gradient is more significant near the edge of the nanoparticles. This assumption is confirmed by our simulation shown in Fig. S4, where the optical gradient force is ~7 times smaller at the center than at the end of the nanorod based on the optical gradient force simulation using Maxwell stress tensor. The distribution of the optical gradient force is similar to the results in Ref. 51.

We have included these assumptions and discussion on page 8 as “We define the amplitude distribution of the optical forces as $F_{opt}(\vec{r})$ and phase as $\varphi_{opt}(\vec{r})$. To delineate these optical forces, we first subtract the non-localized photoacoustic force F_{PA} and its phase φ_{PA} in the field of view based on the background signal outside the nanorod. We assume the background is mainly from the photoacoustic force, and thus it has a phase of 90 degrees. We find the phase of the photothermal force φ from the region with a low contribution of the optical gradient force (i.e., the center of the nanorod). When the photoacoustic force is removed from the overall optical forces, the photothermal force is the only one that contributes to the imaginary component of the remaining optical forces $(F_{opt}(\vec{r}) - F_{PA})\sin(\varphi_{opt}(\vec{r}) - \varphi_{PA} + \pi / 2)$. Thus, the photothermal force can be calculated as $F_{PT}(\vec{r}) = \frac{(F_{opt}(\vec{r}) - F_{PA})\sin(\varphi_{opt}(\vec{r}) - \varphi_{PA} + \pi / 2)}{\sin(\varphi_{PT})}$. The optical gradient force only has a real component in our hypothesis and is decoupled by subtracting the real component of the photothermal force, $F_G(\vec{r}) = (F_{opt}(\vec{r}) - F_{PA})\cos(\varphi_{opt}(\vec{r}) - \varphi_{PA} + \pi) - F_{PT}(\vec{r})\cos(\varphi_{PT})$.”

10. The title and conclusion suggest that the measurements are done with a time resolution of nanoseconds, which is definitely not the case. Variation of the lock-in frequency indirectly informs on the time-response of the different force components, but this analysis cannot replace a time-resolved investigation with pulses. This should be honestly acknowledged and explained.

Authors: we agree with the reviewer that this measurement cannot completely replace a time-resolved measurement and we do not claim our approach will replace all time-resolved measurements. But we believe our approach will offer an additional valuable method to study optical, thermal, or mechanical properties of nanomaterials for physicists, chemists, and material scientists. Additionally, we want to point out that our measurement indeed shows the temperature map of a repeatedly heated gold nanorod in nanometer spatial resolution and the time evolution of the temperature profile in nanosecond temporal resolution as we reported in figure 4. To the best of our knowledge, this is the first approach to reach such a spatial-temporal resolution.

To clarify the difference between our technique and conventional time-resolved measurement methods, such as a pump-probe measurement, we have included additional discussions on page 13, the revised paragraph reads “*This technology can be extended to any samples that follow three criteria: first, the sample can be measured using AFM; second, the photothermal effect is relatively strong, i.e., the sample has a significant light absorption; and third, the thermal relaxation is in a similar time scale to the cantilever’s mechanical resonance. Examples of these samples include polymers, 2D materials, cells, nanoparticles, and macromolecules. As the time-resolving capability of this technique requires a repetitive modulation of the laser, it cannot observe single-shot or random photothermal processes. Further, our measured time-frame-resolving signal is directly related to the photothermal expansion, it cannot replace traditional pump-probe techniques which target ultrafast processes related to general light-matter interactions* ³⁸.”

11. Supplementary Information:

- I think the COMSOL simulations should include elastic deformation in addition to temperature.

Authors: we thank the reviewer for this suggestion. Per the reviewer’s suggestion, we have included elastic deformation in the Supplementary Information. It is now listed in Supplementary Note 3 and Supplementary Fig. S6.

12. Why is the sample limited to 50 nm radius? The heat diffusion continues on larger scales, yielding relaxation over times that grow as the square of the length scale. The effect of previous illumination cannot be ignored.

Authors: we thank the reviewer for pointing out this confusion. The sample is *not limited* to a 50 nm radius. In our simulation, we consider a sufficiently large domain with a length and width of 1 μm and thicknesses of 200 nm and 500 nm for PMMA and glass, respectively. Because PMMA has a relatively low thermal conductivity (0.0374 W/mK), the temperature is not uniform across the domain. We show several temperatures at different locations on the PMMA substrate in Figure R2 below. We previously chose a 50 nm hemisphere to evaluate the temperature of PMMA around the nanorod in Fig. S5a. This parameter is only chosen to interpret the data and will not influence the simulation result. To avoid confusion to the readers, we removed the temperature of PMMA in the SI because it doesn't contribute to our explanation.

Fig. R2| Temperature versus time at different locations of the PMMA substrate. a, Position of the five measurement points. b, Elevated temperature of the five measurement points.

13. The modeling is done with a single light pulse, whereas the actual experiments are performed with a square wave. The accumulation of thermal energy in the sample is not properly accounted for.

Authors: We thank the reviewer for raising the question about a single pulse simulation. Our simulation shows that the temperature of the nanorod and its surrounding region is cooled enough (to below 0.005 K) after one modulation cycle, therefore the simulation of a single period is sufficient. In addition, we measured the AC component of the force through the cantilever's oscillation, the base temperature, i.e., the DC component of the elevated temperature, will not influence the result.

We included the description in Supplementary Note 3 that “ f_{opt} is around 170 kHz, which gives a modulation period of 5.88 μs . As a result, the laser off-time is 4.41 μs . As shown in Fig. S5a, the elevated temperature decreases to be $<0.005\text{K}$ in 2.21 μs after the laser is off. As the elevated temperature of the nanorod drops to be significantly lower than its peak value of 10.9 K, simulation of a single period is sufficient.”

14. The expression for the gradient force is derived from that for a nanoparticle much smaller than the wavelength. There is not guarantee that it applies to the AFM tip with and that the force is independent of the tip position as assumed here. What is E_z ?

Authors: we thank the reviewer for the comment. We have included a full-wave simulation in Supplementary Note 2. As the simulation in Fig. S3 shows, only a small proportion (<25 nm in diameter) of the tip is involved in the polarization, which is much smaller than the wavelength. Therefore, the dipole approximation for nanoparticles can still hold true. E_z is the electrical field in the z-direction, as described on page 5 of the main text.

15. Note 8: the time 32.7 ns is an uncertainty in phase, not a time resolution! The method cannot record a time trace on a time scale of nanoseconds.

Authors: As we have discussed in the manuscript, the time frame sensitivity comes from the back-action of the photothermal expansion. One can choose the time frame by controlling the phase shift between the modulations of the laser and the tip-sample interaction that indicates the photothermal expansion. Therefore, the uncertainty in phase corresponds to our smallest resolvable time separation. Our experimentally measured photothermal force maps in Figure 4 of the main text also show that our temporal sensitivity is indeed at the nanosecond level.

Reviewer #2:

Rev 2: The authors report a method to map the photothermal forces with a spatial resolution at nanometer scale by using optical force nanoscopy. They show the phase responses of the photothermal force under temporal modulation of light. The manuscript reports a further interesting effect where the back-action of the photothermal effect can be used to obtain the dynamical photothermal process of a single gold nanorod in the nanosecond regime. The method developed in this manuscript is new and of broad interests to optics community.

The manuscript is written well with figures of high quality and good structure. The results are interesting, and experimental and simulated data support the conclusions. I would recommend publication of this work in Nature Communications. I suggest that the authors consider the following comments in their final submission:

Authors: we thank the reviewer for highlighting the novelty and broad interest of our research! We appreciate for recommending our manuscript! We have addressed all the comments of the reviewer with the point-by-point response shown below.

Detailed remarks (Reviewer 2):

1. The photothermal effect is usually considered as a specialized effect. It is worth to clearly address its suitability for publication in Nature Communications instead of a more focused journal in regards of the research interest to broad scientific community.

Authors: we thank the reviewer for suggesting elaborating on the broader impact of our work. We have rewritten our introduction to highlight the wide applications of our optical force technique, in addition to the photothermal effect. This technique can be applied as a new characterization tool and study various types of samples and probe their optical, thermal, and mechanical properties.

“Optical forces, also known as light-induced forces, refer to the mechanical effects generated by the interactions between light and matter. When light is absorbed or scattered by a material, it can create non-uniform distributions of the electromagnetic or thermal fields, resulting in mechanical forces due to the transfer of momentum. These optical forces have been widely used in a variety of applications, including trapping^{17,18}, sensing¹⁹, micromanipulation^{20,21}, and surface characterization²².

The measurements of microscopic optical forces have been accomplished using techniques such as atomic force microscopy (AFM)^{23,24}, photo-induced force microscopy (PiFM)^{22,25,26}, and far-field scattering methods^{27,28}. However, the interpretation of these measurements has been complicated by the fact that optical force is an umbrella term encompassing a wide range of forces generated by light-matter interactions. Decoupling these forces is crucial for understanding the optical, thermal, and mechanical properties of materials and for developing efficient sensing, imaging, trapping, and actuating schemes. Despite its importance, decoupling optical forces remains a significant challenge due to the complex nature of light-matter interactions. New techniques and theoretical models that accurately measure and decouple different types of optical forces could lead to advancements in our understanding of light-matter interactions and the development of novel applications in a wide range of fields as diverse as nanophotonics, biophysics, and materials science.

While scanning probe-based force measurement generally provides a high spatial resolution of optical forces; three forces — optical gradient force, photothermal force, and photoacoustic force — are tangled and measured by the probe simultaneously. An approach using PiFM to decouple the optical gradient force from the photothermal expansion using various thicknesses of polymer films is documented⁸; however, it relies on sample thickness and is not applicable to nanomaterials or nanostructures. Additionally, in some cases, optical forces can be transient but have not been measured directly at the nanoscale. The dynamic information could be lost due to the relatively slow scanning speed of the probe. Although improved mechanical designs of the scanner have enabled a high-speed atomic force microscope (Hs-AFM) that can reach up to 1300 frames per second, a single image -frame still takes hundreds of microseconds, significantly longer than the thermal relaxation time of nanoparticles, which is in the nanosecond regime.

Here, we develop a decoupled optical force nanoscopy (Dofn) that can map the optical forces, capitalizing on the unique phase responses of the different optical force components under a specific temporal modulation profile of light. We measured the spatial distribution of these piconewton-level optical force components generated from a single gold nanoparticle with 10 nm resolution. We further demonstrate an ultrafast visualization of dynamic heat transfer in the nanosecond temporal regime using the back-action effect²⁹. We show the heating and cooling stages of the gold nanoparticle using Dofn. Our method provides a promising solution to the long-standing challenge of measuring the fast dynamics of force evolution at the nanoscale.”

2. The example test mass in this manuscript is a gold nanorod. Can authors explain what those test masses are designed for? Why it is essential to map the photothermal force for those substrates? What are the possible applications?

Authors: we thank the reviewer for the question regarding the choice of the sample. We chose gold nanorods as our imaging object due to their promising biomedical and energy applications, including cancer imaging and therapy, neuroscience, as well as their applications in water splitting and catalysis.

From the basic science aspect, gold nanoparticles also involve a complex interplay between all the components composing the optical forces. Therefore, we chose a gold nanorod as a representative sample. Please note that our technique can also study other types of nanomaterials, including various light-sensitive nanoparticles, two-dimensional materials, polymers, and biological specimens. PMMA is chosen as the substrate due to its similar properties to biological environments, such as adipose tissue, relevant to biomedical applications. Furthermore, it shows good thermal resistivity and a high thermal expansion coefficient, both beneficial for visualizing the photothermal force.

3. The main experimental results are demonstrated in a single nanorod. Can authors comment on if this technique can be applied to a target of different materials and dimensions? Is there a limitation?

Authors: Yes, this method is general to many types of samples. These samples can include polymers, 2D materials, nanoparticles, and biological samples. The imaging technique is based on the general optical forces, thus can be applicable for characterizing the optical, thermal, and mechanical properties of optical absorbing nanomaterials. We have included an additional discussion paragraph in the main text on page 12. The technique is suitable to samples as long as (1) the sample is compatible with AFM; (2) the sample has a strong interaction with light; and (3) the thermal relaxation is in a time scale comparable with the cantilever's mechanical resonance, which can cover a wide range from several kHz to MHz. We have revised our manuscript to include these discussions on page 12, which reads *“This technology can be extended to any sample that follows three criteria: first, the sample is compatible with AFM; second, the sample has a strong interaction with light; and third, to measure the photothermal force temporal profile, the thermal relaxation is in a time scale comparable with the cantilever's mechanical resonance.”* There are also limitations to this technique. To reliably measure the temporal evolution of the events, our method requires the optical force to be repetitive following the optical modulation during the scanning, which means this method does not measure a single-shot event.

4. It is not very clear how the optical pulse is generated and if the pulse window has a rise-up and rise-down response that adds an error to the photothermal response time.

Authors: We thank the reviewer for the very good question. The modulation of the laser is performed through an acousto-optic modulator (Gooch & Housego AOMO 3080-125). According to its datasheet, the rise and fall time is between 23 ns and 65 ns, which is comparably small to the thermal relaxation time of 283 ns. We have added more details in our Supplementary Note 1 to clarify the possible errors due to rise-up and rise-down responses. The revised Supplementary Note 1 reads, *“The AOM has a rise and fall time of between 23 ns and 65 ns, which is comparably small to the thermal relaxation time of 283 ns. The slightly non-sharp rising and falling edges will result in a minor phase increase for both the optical gradient force and the photothermal force. However, as both of the optical forces will experience such a phase change, the relative phase relationship between them will be mostly unchanged.”*

Reviewer #3:

Rev 3: This study aims to separate different photoinduced forces between an AFM tip and a plasmonic nanoparticle. They present a method to separate the thermal expansion contribution and the optical gradient (dipole-dipole). The origin of optically-induced tip-sample forces has been the subject of many recent studies and the clarification and separation of these contributions would be highly impactful. However, several items must be clarified and explained in this study for it to be publishable. I have outlined my concerns below.

Authors: we thank the reviewer for recognizing the high impact of our work! We appreciate the constructive comments that the reviewer proposed to improve our manuscript. We have included additional experiments and simulations per the reviewer's comments, and have addressed the reviewer's comments in detail as shown below.

Detailed remarks (Reviewer 3):

1. The assignment of the real part of the optical force to the optical gradient force is not consistent with recent literature which show that the optical gradient force is much weaker than the induced change in van der Waals force due to thermal expansion (Jahng, et al., *Anal. Chem.* 2018, 90, 18, 11054–11061). This mechanism presented in Jahng, et al. also explains the absorptive lineshape of the force spectrum, whereas the spectrum of the optical gradient force should follow the real part of the sample dielectric function (i.e. dispersive spectral profile). The authors should explain why the spectrum they observe is absorptive and not dispersive as predicted previously. The authors should also clarify what optical force contribution is calculated and plotted in Fig. 2k.

Authors: we thank the reviewer for pointing out the inconsistency with the reported result in the literature. The optical force result shown in *Jahng, et al., Anal. Chem.* 2018, 90, 18, 11054–11061 is from the interaction between the AFM tip with a dielectric thin film. Differently, in our paper, the interaction between the plasmonic nanoparticle with the AFM tip will be more significant due to the strong electrical near-field raised by the surface plasmon resonance. The dominant effect in generating the optical gradient force is different for dielectric samples and plasmonic samples. For the dielectric samples, the force is mainly given by the polarization of the AFM tip and its mirror dipole moments on the substrate, which is related to the dielectric properties of the substrate. On the other hand, for plasmonic samples, the polarizability of the sample is much higher than the AFM tip. Therefore, the optical force is mainly given by the localized electric field strength induced by the sample.

To better explain this effect, we have included the discussion above to **Supplementary Note 2. Optical gradient force simulation.** Within the section, we highlight that “*The dominant effect in generating the optical gradient force differs for dielectric samples¹ and plasmonic samples. For the dielectric samples, the force is mainly given by the lightning rod effect of the AFM tip apex and its induced-dipole moments on the substrate that is related to the dielectric properties of the substrate. On the other hand, for plasmonic samples, the polarizability of the sample is much higher than the AFM tip. Therefore, the optical force is mainly given by the localized electric field strength induced by the sample.*”

¹ *Jahng, et al., Anal. Chem.* 2018, 90, 18, 11054–11061

We also add “*The simulated optical force spectra at the end and the center of the nanorod are given by the optical gradient force and the photothermal force, respectively.*” to the caption of Figure 2.

2. Regarding Fig. 3, the authors claim that the resonance is shifted due to back-action but how do they rule out heating and softening of the cantilever with increasing DC laser power? This would have the same effect as the data shown: f_0 would decrease and the linewidth would broaden. The authors would need to measure the cantilever resonance curves for these power levels and rule out that this effect.

Authors: we thank the reviewer for the suggestions of ruling out heating and softening of the cantilever with the increasing DC laser power. This is a very good suggestion. To rule out this effect, we conducted additional experiments by measuring the resonance frequencies while changing the DC laser power. We used a similar laser power as in Figure 3a and a clear glass slide as the sample. We measured the tuning curve of the cantilever as we increased the DC laser power and recorded the shift of the resonance frequency. Our new experiments show that the shift of the resonance due to laser power is within 0.032

kHz, which is one-order of magnitude smaller than the shift induced by back-action. Therefore, the frequency shift due to DC laser power is negligible. We have included the details of these experiments and results in Supplementary Note 11 and Supplementary Figure S15, as cited below.

“Supplementary Note 11: Thermal softening of the cantilever

To rule out the possibility of softening of the cantilever due to increasing DC laser power, we conducted additional experiments by measuring the resonance frequencies while changing the DC laser power. Our experiments confirm that the shifted resonance is indeed from the back-action of the photothermal expansion instead of the heating and softening of the cantilever. As shown in Fig. S15a below, we used a similar laser power as in Figure 3a and a clear glass slide as the sample. We measured the tuning curve of the cantilever as we increased the DC laser power and recorded the shift of the resonance frequency. Our new experiment shows that the shift of the resonance by such an effect is within 0.032 kHz.

We compare the experimental results when the tip is engaged to a nanorod (Fig. S15b). Fig. S15c shows that the shift of resonance due to the heating and softening of the cantilever is around one order of magnitude smaller and also in opposite direction compared to the shift due to back-action. The significant difference in the frequency shift suggests that the changes in resonance in Dofn are mainly due to the back-action of the photothermal expansion, whereas the frequency shift due to the DC laser power change is negligible.

Fig. S15| Shift of resonance frequency by heating and softening of the cantilever and back-action of the photothermal force. f_0 denotes the mechanical resonance frequency of the cantilever without laser. **a**, Tuning curve with different DC laser intensity. **b**, Oscillation at the laser modulation frequency with different AC laser intensities. **c**, The comparison of frequency shift at different laser intensities of the two cases.”

3. The authors’ procedure to compensate drift in their images is not appropriate. They should present the raw, uncorrected data, or omit the data points lost due to drift. The method of moving data points from one side of the image to another cannot be used.

Authors: We thank the reviewer's suggestion on data presentation. Accordingly, in the revised manuscript, we have updated the main text Figures 2 and 4 by omitting the data points lost due to drift. We have included the raw data before correcting the drift and omitting the data points in Supplementary Note 10. We also updated the Fig. S9 in Supplementary Note 5 to reflect this change.

“Supplementary Note 10: Correction to the thermal drifting

We recorded the amplitude and phase of the optical force while scanning the sample (Fig. S14a and S14b). Each recording of a force map typically takes around 4 minutes. The nanorod will drift around 50 nm due to the thermal drift of the sample and/or the probe. This drift can be corrected with post-processing assuming a constant drifting velocity. As shown in Fig. S14c, after drift-correction, the nanorod shifts back to a vertical position. We then omitted the background data that exceeds the rectangular boundary in Figure S14, as well as Figures 2 and 4 in the main text.”

Supplementary Fig. S14| Correction of the thermal drift of the scanning probe. Raw data of **a**, amplitude and **b**, phase of the optical force. Map of the **c**, amplitude and **d**, phase of the optical force before drift correction. Map of **e**, amplitude and **f**, phase of the optical force after drift correction.

4. I don't understand how the authors measure the force at different time delays (Fig. 4). Detuning the laser repetition rate off of resonance would not probe the same time delay at each pulse, it would only change how effectively the force excites the tip resonance. On this note, the authors should state what excitation frequency and the tip tapping frequency was used for the measurement in Fig. 2.

Authors: we appreciate the reviewer for pointing out such a confusion. Per the reviewer's suggestion, we have included that "*The laser modulation, f_{opt} , is at the same frequency as the mechanical resonance frequency of the cantilever, f_0 . The frequency difference $f_d - f_0$ is 1 kHz.*" in the caption of Figure 2.

We did not change the laser modulation frequency, f_{opt} . To probe different time frame, we change the piezo's modulation frequency and fix the engagement factor, consequently, we change the phase-shift between the laser's modulation and the back-action of the photothermal expansion. As discussed in Figure 3, the phase shift decides how the tip-sample interaction is shifted to the laser modulation, therefore, decides the time frame being probed. The time delay corresponds to the phase through the equation on page 11, $t_{probe} = t_0 + \frac{\phi}{f_{opt}}$. Please note that the phase ϕ , as the phase shift between the laser modulation and the back-action of the photothermal expansion, is different from the phase of the optical force.

To avoid this confusion, we include that "*The probed time frame can be tuned by the phase shift between the back-action of the photothermal expansion and the laser modulation, ϕ , as $t_{probe} = t_0 + \frac{\phi}{f_{opt}}$, where t_0 is a constant given by the initial condition. Please note that that the phase we show in Figure 3, as the phase involved in the tip-sample interaction, is different from the phase of the optical force. ϕ contains temporal information of the photothermal expansion, while φ_{opt} contains physical information of the optical force's origins.*" to page 11 of the manuscript.

5. How do the authors account for the shift in the tapping phase during imaging? Unless the laser is well synchronized with the motion of the tip, changes in sample viscoelastic properties could leak into the force phase channel, especially at edges in the topography. Did the authors also measure the AFM phase image?

Authors: this is an excellent question! The shifting of the tapping phase will leak into the force phase channel. This is unavoidable. However, please note that our laser is modulated at a different frequency than the piezo's modulation, as a result, such an influence would be minor. We indeed measured the AFM phase image. As evidence, we have included the phase image of the AFM scan in Fig. R3 below. It is seen that the phase image of the AFM scan is significantly different from the phase map of the optical force in Figure 2c of the main text. Specifically, the phase map in Fig. 4b does not show the two "hot spots" at the two ends of the nanorod in Figure 2c, which is induced by the optical gradient force. The phase in Fig. 4b is higher around the nanorod but is flatter in the optical force map in Figure 2c.

Fig. R3| Topography and phase of an AFM scan. a, Topography of AFM scan of a gold nanorod. **b,** Phase of AFM scan of a gold nanorod.

6. Minor concerns:

The integral in Eq. 2 is unclear. What distance is the integral calculated over?

Authors: the integral is performed in the area where the elevated temperature is significant, which is within 500 nm. Per the reviewer’s suggestion, we have included such detail in Fig. S6c of **Supplementary Note 3. Photothermal simulation.**

“The photothermal force is generated by the thermal expansion around the nanorod. In equation (2) of the main text, we simplify such an expansion as an isotropic displacement along the probing direction (dashed line along the z-direction as shown in Fig S6a). Due to the irregular shape of the sample and the AFM tip, such a simplification may yield an error compared to the simulated photothermal expansion-induced displacement of the cantilever. As shown in Fig. S6a, we simulate the thermal expansion induced by the heated nanorod with an elevated temperature of 10.9 K (close to the one shown in Figure 4a in the main text). Because the size of the nanorod (90 nm in length and 30 nm in width) is over three orders of magnitudes smaller than the AFM probe (170 μm in length and 40 μm in width), meshing the entire domain to achieve an accurate calculation will cause a significant computational burden. To reduce the computational burden and the meshing difficulty, we increase the size of the nanorod by 10 times but keep the temperature elevation the same as our previous calculation based on the actual nanorod dimension. We simulate the displacement of the cantilever using COMSOL Solid Mechanics Module (Fig. S6b). The surrounding 4 surfaces of the substrate and the base of the cantilever are set as fixed boundaries. The temperature distribution along the probing direction is shown in Fig. S6c.

We calculate the displacement with equation (2) and compare it with the simulated displacement (Fig. S6d). The simulated displacement of the cantilever from the thermal expansion shows a difference from the calculated one given by equation (2) by less than 20%. The difference mainly comes from the non-isotropic expansion of the tip and the sample along the z-direction and would be smaller for the actual size because the heat will be spatially more confined. The spring constant of the cantilever was measured to be 8.6 N/m. By multiplying the spring constant and diving by the scale factor of 10, we get an approximate photothermal force of 70.8 pN, which roughly fits our measurement. The slightly bigger value from the theoretical calculation may be attributed to the less confined temperature distribution because we used a larger rod in the simulation compared to the experiment.”

Supplementary Fig S6| Multiphysics simulation of the photothermal expansion. *a*, Simulated elevated temperature distribution. The elevated temperature of the nanorod is set as 10.9 K based on the simulation in Fig. S5. We increase the nanorod by 10 times to a length of 0.9 μm , and a width of 0.3 μm , and simulate the substrate with a 50- μm -thick glass layer. *b*, Simulated displacement distribution. The probing direction is marked as the dashed line. *c*, Temperature distribution along the probing direction. The subplot shows the position of the substrate ①, nanorod ②, and AFM tip ③. *d*, Theoretical and simulated thermal expansion along the probing line. The theoretical curve is calculated by equation (2).”

7. There are many grammatical errors and confusing phrases throughout the manuscript. I recommend a careful proof-reading before resubmission.

Authors: we thank the reviewer for addressing the grammar errors and confusing phrases. We have had a native speaker to read through the manuscript and carefully corrected the language errors. We have highlighted changes in the following.

Page 2, “in nanoscale” change to “at the nanoscale”; change “Light and plasmonic nanoparticle interaction creates many physical effects” to “Light and plasmonic nanoparticle interactions create many physical effects”.

Page 2, add “is” before “diffraction limited”.

Page 3, change “the dynamical information can be lost” to “the dynamic information could be lost”; change “originated from” to “originating from”.

Page 3, change “super continuum laser” to “supercontinuum laser”; change “build-in” to “built-in”.

Page 8, change “in the most of” to “in most of”.

In the caption of Figure 1a change “schematics” to “schematic”.

In the caption of Figure 1a change “schematics” to “schematic”; change “response” to “responds”.

Reviewer #1 (Remarks to the Author):

In their revised manuscript, the authors clarify some of my questions. However, despite their lengthy reply, some other points are left open.

- the difference with reference 8 still seems incremental to me. As I understand it, the essential change is one of the modulation frequency. The main novelty is in the analysis method, which by itself does not justify publication in a high-impact journal.

- optical force: I agree that the origin of optical forces is now explained convincingly with simulations of the Maxwell tensor.

- photothermal force: the origin of the "photothermal force" is indeed the change of distance to the substrate due to thermal expansion. However, why call it a force, when it follows from a change of displacement? The linear relation between displacement and force only applies if the dithering is very small compared to the range of the tip-substrate interaction. It does not apply in the tapping mode.

- photoacoustic force: I understand and agree that viscosity forces from the air are presumably negligible. Yet, I do not see why any interactions would be well represented by the pressure of acoustic waves in a regime where distances (in the nanometer range) are much shorter than acoustic wavelengths (in the micrometer range). What is the validity of acoustic waves on scales which are not only shorter than the wavelength, but probably even shorter than the mean free path of air molecules?

- I still have doubts about phase when the modulation is square instead of sinusoidal, because the different harmonics of the square wave will have different phases.

Upon the responses to my specific points:

- point 1: the response is not relevant, as the scales are much larger.

- point 2: I agree with the new supplementary note, which clarifies the simulation of the optical force, but what is reference 53?

- point 3: I am not sure to understand the authors' answer and arguments. Are continuous fluid equations valid when applied to a gas at nanoscales where the mean free path is around 100 nm?

- point 8: I maintain that thermal relaxation is non-exponential, as heat diffusion in a continuous medium occurs upon many length scales, each one involving a different time scale (proportional to the square of the length scale). This deviation is clearly visible in Fig. R1, where the exponential decay is clearly faster than the simulated points. Under such conditions, a decay time as precise as 282.9 ns (with which inaccuracy?) has no physical meaning.

In conclusion, I reiterate that the presented data are potentially interesting for specialists of AFM imaging, who will themselves supplement the missing information. For a general readership of Nature Communications, however, I still find the paper very unclear and making too many tacit assumptions. The incremental novelty of the reported results does not, in my view, justify publication in Nature Communications.

Reviewer #2 (Remarks to the Author):

After revision, the authors have addressed all the concerns I had. I would like to recommend publication of this manuscript at nature communications.

Reviewer #3 (Remarks to the Author):

The authors have adequately addressed my concerns and the manuscript is suitable for publication.

Authors' Responses to the Reviewers' Comments on manuscript NCOMMS-22-36661A "Visualizing Ultrafast Photothermal Dynamics with Decoupled Optical Force Nanoscopy"

We are thrilled that reviewer 2 and reviewer 3 agree that our manuscript is suitable for publication. We thank the constructive comments from reviewer 1. In the following, we address the additional comments from reviewer 1 point by point. We also made the changes accordingly in our manuscript.

Reviewer #1:

Reviewer: In their revised manuscript, the authors clarify some of my questions. However, despite their lengthy reply, some other points are left open.

Authors: We thank the reviewer for taking the time to review our response and our revised manuscript. We are glad that the reviewer thinks that our revision has clarified some of his/her questions. We are happy to answer the remaining questions and address any concerns that the reviewer brought up.

Reviewer: the difference with reference 8 still seems incremental to me. As I understand it, the essential change is one of the modulation frequency. The main novelty is in the analysis method, which by itself does not justify publication in a high-impact journal.

Authors: We respectfully disagree that our contribution is incremental. There is a conceptual difference between our method and the method reported in Ref. 8. Although Ref. 8 discusses the multiple origins of the forces, their method cannot measure them separately, let alone capture the dynamic photothermal effect. In Ref. 8, the authors reported that different magnitudes of the mixed forces were observed with different thicknesses of the thin film. They suggested that the substrate thickness contributed to the change of photothermal force, thus causing the change in overall mixed force in the measurement. Although Ref 8 presents an interesting observation and result, there is no direct demonstration of the optical force dissociation in their results.

Additionally, compared to large-scale thin film samples, nanoscale structures, and nanomaterials are much more challenging to measure and interesting to study, but currently, there are limited alternative methods for measuring the optical forces from these samples, and none of the reported approaches can differentiate the optical force components or at the same time provide nanosecond dynamic information as we have presented in this study. Thus, our contributions are significant because our method represents a general approach to characterizing nanophotonic devices, optical nanomaterials, as well as thin films.

To highlight our novelty, we have further revised our conclusion paragraph to include the general applicability of our method. We rephrase "*The capability to decouple optical force and visualize the ultrafast photothermal processes from a single nanoparticle will enable new ways to develop efficient photothermal nano-agents*" to "*Our technique adeptly decouples various optical force components and captures the temporal dynamics of photothermal forces. Measuring nanoparticles paves the way for crafting effective photothermal nano-agents. For instance, by tweaking the nanoparticle's material composition, we can discern distinct photothermal distributions.*"

We also expanded the applications as "*Moreover, Dofn enables the spectroscopic analysis of emerging optical nanomaterials with unmatched spatiotemporal resolution, aiding in the characterization of nanophotonic devices, the design of nanoscale optical traps, and the innovation of thermal photonic systems.*"

Reviewer: optical force: I agree that the origin of optical forces is now explained convincingly with simulations of the Maxwell tensor.

Authors: We thank the reviewer for acknowledging our convincing results.

Reviewer: photothermal force: the origin of the “photothermal force” is indeed the change of distance to the substrate due to thermal expansion. However, why call it a force, when it follows from a change of displacement? The linear relation between displacement and force only applies if the dithering is very small compared to the range of the tip-substrate interaction. It does not apply in the tapping mode.

Authors: We thank the reviewer for confirming that the photothermal force is from the thermal expansion.

We do not disagree that photothermal expansion is a displacement, while the displacement of thermal expansion interacts with the cantilever as a force. We would like to point out that we are not the first ones who termed it photothermal force. For example, David Woolf, Pui-Chuen Hui, Eiji Iwase, Mughees Khan, Alejandro W. Rodriguez, Parag Deotare, Irfan Bulu, Steven G. Johnson, Federico Capasso, and Marko Loncar, “Optomechanical and photothermal interactions in suspended photonic crystal membranes”, *Optics Express* 21, 7258-7275 (2013), described “Actuation in our system is achieved by a combination of optical and photothermal forces, the latter of which arises from absorption of light in the cavity that generates displacements through thermal expansion at the Silicon-SiO₂ interface.” The terminology of “photothermal force” can be also found in the context of scanning probe measurement, such as Ref. 26.

We are pleased to note that the reviewer concurs with the suitability of the tapping mode for measuring photothermal expansion, particularly in cases of minimal dithering. In certain literature, this mode is referred to as the non-contact mode when the tip operates within the attractive regime. This condition reflects our situation. As an example, this approach was employed in Ref. 26, where the authors utilized a free-air amplitude of 10 nm alongside an 88% setpoint, resulting in an oscillation amplitude of 8.8 nm. They denoted it as tapping mode/non-contact mode. In our study, we have chosen to refer to it simply as tapping mode, as the cantilever oscillates, distinguishing it from the static condition (contact mode). In parallel with this, we have employed a similar magnitude of dithering amplitude as Ref. 26. To be precise, our approach utilizes a free air amplitude of 15.9 nm, with a setpoint of 70%, yielding an oscillation amplitude of 11.1 nm—akin to the order of magnitude outlined in the literature. We have incorporated this experimental detail into the revised caption of Figure 4: “*The free air amplitude is 100 mV, corresponding to 15.9 nm. The measurement is conducted with a setpoint of 70%.*”

In addition, our photothermal expansion is modulated at a different frequency from the piezo’s dithering frequency. We chose this intentionally because the modulated photothermal expansion can drive a distinct oscillation of the cantilever through the tip-sample interaction, allowing us to extract both the amplitude and phase. The distinct oscillation at the optical modulation frequency is detected with the lock-in amplifier, where the amplitude reflects how strongly the photothermal force drives the cantilever. We discussed how and why these frequencies are chosen in detail in “Supplementary Note 6: Observation of the back-action of the photothermal expansion” and “Supplementary Note 7: Theory of phase tunability of the Dofn system”, and “Supplementary Note 9: Sensitivity of the optical force measurement”.

Reviewer: photoacoustic force: I understand and agree that viscosity forces from the air are presumably negligible. Yet, I do not see why any interactions would be well represented by the pressure of acoustic waves in a regime where distances (in the nanometer range) are much shorter than acoustic wavelengths (in the micrometer range). What is the validity of acoustic waves on scales which are not only shorter than the wavelength, but probably even shorter than the mean free path of air molecules?

Authors: The photoacoustic wavelength is in the micrometer range, which aligns with our analysis and discussion. In the main text and Supplementary Note 4, we mentioned that the photoacoustic force is not mainly probed by the AFM tip but by the entire cantilever. The cantilever has a size of around 175 μm , which is the reason that the photoacoustic acoustic force can be treated as the background force, as shown in Supplementary Fig. S7.

Here we cite our discussion on page 7 of the main text “*The pulse length in air is, therefore, around 160.1 μm . It is on the same length scale as the cantilever’s size of around 175 μm but much larger than the tip*”

size of around 20 nm. As a result, the photoacoustic pressure will exert on the entire probe instead of a localized spot near the tip”. On page 8 of the main text, we also mentioned that “... the non-localized photoacoustic effect dominates the background signal but contributes minimally to the structured distribution around the gold nanorod”. To further improve clarity, in this revised manuscript, we have added a phrase after the entire probe as “the entire probe (*i.e.*, cantilever and tip)”.

Reviewer: I still have doubts about phase when the modulation is square instead of sinusoidal, because the different harmonics of the square wave will have different phases.

Authors: It is true that the different harmonics of the square wave exhibit different phases, however, it’s important to note that our system does not measure the higher-order harmonics. This is primarily due to two reasons: Firstly, the lock-in amplifier measures the fundamental harmonic, and secondly, the higher harmonics are off-resonance of the cantilever.

Our lock-in amplifier uses a reference frequency the same as the fundamental harmonic of the square wave. We set our lock-in amplifier’s low-pass filter bandwidth to 10 Hz, and the high-frequency components, including the higher-order harmonics, have been removed. We have incorporated the lock-in reference frequency information on page 20 of the revised main text in the Methods section, stating that “The lock-in amplifier utilizes a reference frequency same as the fundamental harmonic of the square wave, therefore, excluding the potential high-order harmonics.”

Furthermore, even in the event of any possible leakage in the low-pass filter, the higher-order harmonics of the optical forces would not drive the cantilever because they will be off-resonance. To elucidate, we present the thermal tuning data of the OPUS 4XC-NN cantilever in Fig. R1 below, which is the same type of cantilever we used in our measurement. The fundamental resonance of this cantilever occurs at 160 kHz, and the second resonance occurs at 385 kHz. Consequently, the higher harmonics from the square wave excitation will not align with the cantilever’s resonant frequencies.

Fig. R1. Thermal tuning data of OPUS 4XC-NN cantilever, showing the fundamental mechanical resonance of 160 kHz, a second resonance of 385 kHz.

Reviewer: Upon the responses to my specific points: point 1: the response is not relevant, as the scales are much larger.

Authors: We are uncertain about the “scales” that the reviewer refers to. We tried our best to make an educated guess that the “scales” the reviewer meant were the targeted sample size. We would like to highlight that conventional pump-probe microscopy based on far-field radiation is diffraction-limited, regardless of the sample size. So far, there is no effective way to perform the pump-probe technique for measuring the photothermal dynamics in ambient conditions with a nanoscopic resolution.

Reviewer: point 2: I agree with the new supplementary note, which clarifies the simulation of the optical force, but what is reference 53?

Authors: We thank the reviewer for pointing out the typo. The reference we were referring to should be *ref. 35* (Juan, M.L., Righini, M. & Quidant, R. Plasmon nano-optical tweezers. *Nature photonics* 5, 349-356 (2011)).

Reviewer: point 3: I am not sure to understand the authors' answer and arguments. Are continuous fluid equations valid when applied to a gas at nanoscales where the mean free path is around 100 nm?

Authors: The fluid equations we referenced in our previous response remain valid in a gaseous medium. The theory for the photoacoustic signal in gases is well established in existing literature, as exemplified by the work of Westervelt and Larson (1973) in their study titled "Laser-excited broadside array" (*The Journal of the Acoustical Society of America*, 54(1), 121-122).

Additionally, despite the mean free path (MFP) of air is ~ 100 nm, this characteristic does not contradict our measurements and theoretical framework. It is crucial to note that the photoacoustic pressure generated from the nanorod propagates outward with a wavelength in the range of microns, which is comparable to the cantilever length, and significantly larger than the MFP of air by orders of magnitude. During the photoacoustic force measurement, not only the tip but also the entire cantilever (175 μm in length) interacts with the photoacoustic pressure. The photoacoustic force in our Dofn setup, thus was measured as the integrated force acting on the cantilever as shown in Equation (7), and Supplementary Figures S7 and S8 in the SI. To clarify, we have added "cantilever and tip" after "the entire probe" in the main text when we described these details, "... *the photoacoustic pulse width, as shown in Figure 1d, is around 471 ns. The pulse length in air is, therefore, around 160.1 μm . It is on the same length scale as the cantilever's size of around 175 μm but much larger than the tip diameter of around 20 nm. As a result, the photoacoustic pressure will exert on the entire probe (i.e., cantilever and tip) instead of a localized spot near the tip, resulting in a non-localized photoacoustic force (Supplementary Note 4), which can be treated as a uniform background.*"

Additionally, we have described the non-localized nature of the measured photoacoustic force in Supplementary Note 4. Specifically, "*Figure S7c shows the photoacoustic force is a uniform background force. In the measurement, the total photoacoustic background force is contributed by an ensemble of nanorods in the illumination area.*" This description further explains that the measured photoacoustic force is not just at the tip area.

Reviewer: point 8: I maintain that thermal relaxation is non-exponential, as heat diffusion in a continuous medium occurs upon many length scales, each one involving a different time scale (proportional to the square of the length scale). This deviation is clearly visible in Fig. R1, where the exponential decay is clearly faster than the simulated points. Under such conditions, a decay time as precise as 282.9 ns (with which inaccuracy?) has no physical meaning.

Authors: We have revised Fig. R1 (now R2) to include the difference between the theoretical calculation and the simulation. As evidenced in the revised Fig. R2, the difference between the theoretical and simulation results is less than $1/30^{\text{th}}$ of the peak temperature elevation. We believe this minor difference is mainly from numerical errors, which corroborate our theory that this process is exponential in our system, thus, it can be characterized by the decay time. An exponential thermal relaxation of plasmonic nanostructures has been reported in the literature, for example, "Chen, Xi, et al. Nanosecond photothermal effects in plasmonic nanostructures. *ACS nano* 6.3 (2012): 2550-2557."

Fig. R2| Simulated temperature compared to the exponential relaxation process.

Reviewer: In conclusion, I reiterate that the presented data are potentially interesting for specialists of AFM imaging, who will themselves supplement the missing information. For a general readership of Nature Communications, however, I still find the paper very unclear and making too many tacit assumptions. The incremental novelty of the reported results does not, in my view, justify publication in Nature Communications.

Authors: We thank the reviewer for highlighting the potential impact of our work. Although we presented an AFM imaging technique, our method is the first one that has demonstrated the physical origins of optical forces from nanoscale samples and shown the nanosecond temporal dynamics of the heating and cooling process of a single nanoparticle for the first time. We anticipate that our method will enable the design and characterization of nanophotonic and thermal-photonic devices and systems, as well as enable the discovery of the fundamental physics of novel optical nanomaterials. We believe the general readers in the fields of nanotechnology, nanophotonics, and materials science would greatly benefit from this new method.

Reviewer #2 (Remarks to the Author):

After revision, the authors have addressed all the concerns I had. I would like to recommend publication of this manuscript at nature communications.

Authors: We thank the reviewer for recommending publication!

Reviewer #3 (Remarks to the Author):

The authors have adequately addressed my concerns and the manuscript is suitable for publication.

Authors: We thank the reviewer for recommending publication!